# LEARNING WITHOUT MEMORIZING CONSIDERED INFEASIBLE: RETHINKING MEMORIZATION IN LLMS

## ABSTRACT

In this paper, we rethink memorization in large language models (LLMs). Memorizing when learning is considered undesirable for two distinct reasons: first, from a privacy perspective, memorization raises concerns about potential leakage of sensitive information in training data. Second, from a learning perspective, memorization raises concerns of sub-optimal learning and over-fitting. We find that existing *measures of memorization*, namely recollection-based and counterfactual measures, are designed to capture privacy concerns, but they ignore optimal learning concerns. We propose a new memorization measure, called *contextual memorization* that captures LLMs tendency to locally over-fit some strings in the training data before during multiple epochs of training.

Applying these measures when training LLMs leads us to two interesting conclusions. First, a systematic analysis of all the measures shows that our new measure avoids a major pitfall of prior measures, by distinguishing context-based recollection from memorization-based recollection of a training string. Using our measure, we revisit prior reported instances of training data memorization by real-world LLMs and find that many instances can be explained away by contextual learning-based recollection, i.e., the prior memorization reports are likely exaggerated. Second, we find that when LLMs learn a language optimally, they inevitably end up *memorizing* some portions of the training data. We support our conclusion with extensive experiments training 18 LLMs from 6 model families to learn a variety of formal languages.

## 1 INTRODUCTION

> *"Every teacher knows that there is a profound difference between a student learning a lesson by rote and learning it with understanding, or meaningfully."* – Herbert Simon

The unsupervised training and fine-tuning of generative models, particularly autoregressive large language models (LLMs), can lead to learning of the training data *by rote* (Bender et al., 2021) and *with understanding* (Bubeck et al., 2023). *Memorization* by rote is considered the ugly cousin of contextual *learning* with understanding; an undesirable side effect of learning that should be avoided (Hernandez et al., 2022). The central question that motivates our work here is *can memorization be avoided when learning?* The answer we find is that **learning a language without memorization is infeasible**. At the same time, we find that **estimates of memorization by LLMs today are likely exaggerated.**

We arrive at our conclusion by re-examining how researchers *operationalize* memorization, i.e., the frameworks they use to understand, measure, and distinguish between the instances when the generation of a string by an LLM is attributed to memorization versus learning. Our contention is that many measures of memorization in use today are quantifying the undesirable effects of memorization rather than the underlying causal phenomenon, i.e., memorization itself.

**Recollection-based Measures.** Privacy researchers, who are concerned about the risks of extracting sensitive information from training data by prompting LLMs, have proposed to estimate memorization by how well LLMs can *recollect* training strings (Schwarzschild et al., 2024; Carlini et al., 2021; Tirumala et al., 2022)[1]. However, there can be cases when such recollection is not based on mem-

---

[1] We provide an extended discussion on existing memorization measures in the Appendix C.

orization. For example, consider asking an LLM to `count from 1 to 1000`. Many LLMs will likely generate $1, 2, \cdots, 1000$ based on simple reasoning. To refer to such generation as *grey area* for memorization, as done in Schwarzschild et al. (2024), risks mis-classifying contextual learning as memorized recollection. In Section 5, we reanalyze strings that prior works have reported as having been memorized by LLMs. We find that most strings are predictable with contextual reasoning and few have privacy sensitive information (that is typically not in public domain). Put differently, *estimates of memorization by LLMs today are greatly exaggerated.*

**The Case for Contextual Measures.** How else could one quantify memorization? Let's first conduct a thought experiment to illustrate a challenging desideratum for memorization measures. Imagine an English speaker and a German speaker commit a paragraph in German to memory. When recollecting the paragraph, do the two speakers rely on memorization to the same or different extents? Intuitively, the German speaker understands the syntax and semantics of the tokens in the paragraph, while the English speaker sees the paragraph as a sequence of alphabet tokens. Even before reading the paragraph, given some prefix, the former is more likely to predict the next token correctly than the latter. So it stands to reason that the extent of memorization involved in recollecting the paragraph is higher for the English speaker than the German speaker. A good memorization measure should account for the ability of *an LLM to predict the next token in a string based on the context.*

We now propose a measure, *contextual memorization*, which can disentangle the effects of context-based recall from those of memorization-based recall. The key intuition, shown in Figure 1, is the following: for each string $s$ in the training dataset $D$, we first estimate its *optimal* contextual recollection, obtained by repeatedly training over a dataset $D'$ that excludes $s$ from $D$ and finding $s$'s best recollection. We declare $s$ as being contextually memorized, if its recollection when included in training exceeds its optimal contextual recollection.

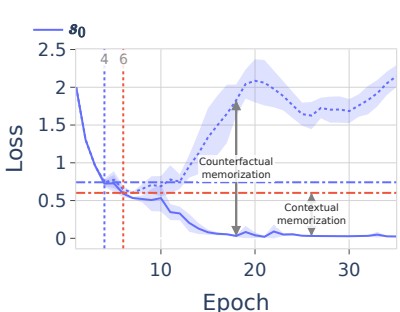

Figure 1: **Contextual Memorization.** Training loss (solid) and contextual loss (dotted) of a representative string $s_0$ from a formal language $L$, when the string is included and excluded from the training dataset $D \sim L$ (detailed in Section 3.2). With repeated training over $D$, the LLM's training loss of $s_0$ decreases along training epochs. In the same way when training on $D' = D \setminus \{s_0\}$, the contextual loss of $s_0$ decreases for a few epochs until it reaches the optimal (i.e., lowest) contextual loss. Thus, the LLM can generate $s_0$ to *some* extent without seeing it during training, but for attaining contextual understanding of $L$. Contextual memorization begins when training loss is lower than the optimal contextual loss (red line), indicating local over-fitting of $s_0$.

**Comparing with Counterfactual Measures.** Contextual memorization differs from the recently proposed *counterfactual memorization* (Zhang et al., 2021), which also relies on comparing recollection of $s$ on training dataset $D$ and dataset $D'$ that excludes $s$, in two subtle but important ways. First, counterfactual measures capture the *per-epoch divergence* in the recollection performance of $s$ over training on $D$ and $D'$, while contextual measures capture the *all-epoch best* recollection performance of $s$ for training on $D'$. Consequently, contextual measures impose a stricter threshold than counterfactual measures (Figure 1). Second, the inspiration for counterfactual measures comes from differential privacy and the potential for inferring the membership of a string $s$ in a training dataset $D$. In contrast, the motivation for contextual measures is rooted in concerns that memorization is an undesirable form of learning, i.e., it represents a type of *local over-fitting* to string $s$ that harms *generalization locally* (van den Burg & Williams, 2021).

**Learning-Memorization Tradeoffs.** Given that memorization is a local phenomenon measured at the level of individual strings $s$ in training dataset $D$ and learning is a global phenomenon measured over a test dataset over some language $L$ from which $D$ is sampled, a natural questions that arises is *can we learn a language $L$ without memorizing any strings $s$ in $L$?* Based on extensive analyzing, using different memorization measures, we conclude that *learning without memorization is infeasible.* The key underlying intuition is the following: every string $s$ in $L$ has its own training epoch $e_s$, when its starts to be memorized and these vary significantly across different strings. The train-

ing epoch $e^*$ corresponding to globally optimal learning often occurs after some (and often many) strings have been memorized.

**Contributions and Implications of our Study.** Our first contributions are our two main findings. One questions if the quest to train LLMs without memorization is an impossible one (Section 4) and the other questions the current assessments of the threat of LLM memorization (Section 5).

The second contribution is our justification of these arguments through a critical re-examination of existing measures of memorization, filling the gaps with new measures, and evaluating them over 18 LLMs across 6 model families and multiple formal languages. We have several key findings that highlight how the precise memorization measure used can impact the determination of when a string $s$ starts to be memorized and to what extent (Section 3).

The third contribution is the controlled setup where an LLM is trained on strings from a formal language. This setting enables precise control over data generation, avoids contamination, and allows manipulation of language entropy to probe the nuances of different memorization measures.

Finally, while memorization mitigation methods like training data deduplication (Kandpal et al., 2022; Lee et al., 2021) are not the main focus of this study, we call for critically re-investigating them (Appendix G). Such methods are increasingly being used to mitigate memorization, as quantified by recollection-based measures. We establish that recollection measures, while easy to use, can lead to misleading conclusions compared to other measures. Therefore, we advise caution against using recollection measures as the target for memorization mitigation, by recalling Goodhart's law that states *when a measure becomes a target, it ceases to be a good measure* (Strathern, 1997).

## 2 ON COMPARING MEASURES OF MEMORIZATION IN LLMS

As the phenomena of memorization arises from an LLM repeatedly training over a dataset, there should be an epoch (iteration) of training when each string in the training dataset begins to be memorized. In subsequent epochs after memorization begins, the extent (measure) of the string's memorization will likely increase till memorization is maximized. Our hypothesis is that comparing how well different memorization measures capture when contextual learning stops and rote learning begins during training would offer us insights into their relative strengths and weaknesses.

In contrast, prior studies proposing memorization measures avoided carefully examining the training dynamics of the model (Schwarzschild et al., 2024; Carlini et al., 2021; Zhang et al., 2021; Carlini et al., 2022). While these measures allowed the studies to determine whether some pre-trained LLM memorized some string without access to training traces, they also overlooked nuanced differences between how the measures evolve over the course of model training. Specifically, we ask the following two questions.

**Formal Setup.** An LLM $M$ is trained on a finite dataset $D$ repeatedly over multiple epochs. $D$ is a random sample of strings from an underlying language $L$, as explained shortly, and may contain duplicated strings. For each string $s \in D$, we wish to answer the following two questions:

- **RQ1 (Memorization Detection Question):** At what epoch $e_s$ does $M$ start to memorize $s$?
- **RQ2 (Memorization Score Question):** What is the degree of memorization or memorization score, $\text{mem}(s, e) \in [0, 1]$, of string $s$ at an epoch $e \geq e_s$? Trivially, $\text{mem}(s, e) = 0$ if $e < e_s$.

We propose to answer **RQ1** and **RQ2** by applying three measures of memorization, as detailed in Section 3. Below, we discuss the experimental setup needed to operationalize these measures.

**Experimental Setup.** We train an LLM on strings from a formal language, focusing on learning syntactic patterns defined by a formal grammar. We choose formal grammar based languages because they offer a controlled setup where we can ensure that learning and memorization are unaffected by prior training of the models, free from data contamination, and guided by a tunable string distribution – enabling detailed comparisons of the memorization measures. While some prior studies have adopted similar setups, their goals differed from ours, such as exploring the representation capabilities of LLMs (Delétang et al., 2022; Bhattamishra et al., 2020) and investigating the difficulty of learning specific languages by certain transformer architectures (Borenstein et al., 2024; Hahn, 2020; Cotterell et al., 2018).

Specifically, we consider probabilistic and hierarchical context-free languages, which mimic the recursive structure of natural language (Allen-Zhu & Li, 2023). Formally, a probabilistic formal language $L$ is defined on a set of allowed tokens or alphabet $T$, and specifies a probability distribution $P_L$ over strings, $P_L : T^* \to [0, 1]$, where $T^*$ is the set of all strings. Throughout, we use the entropy of a language as a key dimension for studying memorization vs. learning. Adjusting entropy alters the frequency of strings, which is a factor central to many memorization measures (Zhang et al., 2021). The entropy $H(L)$ of a language $L$ is the entropy of the probability distribution of strings, $H(L) = -\sum_{s \in T^*} P_L(s) \log P_L(s)$ (Cover, 1999; Carrasco, 1997).

We experiment with 18 open-source LLMs from 6 families, such as Mistral (Jiang et al., 2023), Llama (Dubey et al., 2024), Qwen (Yang et al., 2024), Gemma (Mesnard et al., 2024), Pythia (Biderman et al., 2023), and Opt (Zhang et al., 2022), ranging from 0.5B to 13B parameters. All reported results are averaged over three experimental runs, except in Figure 39 where variance is for different strings. Due to space limit, we defer discussion on formal languages and training details to the Appendix E. Informally, our experiments are based on 8 languages $\{L_1, \ldots, L_8\}$ of varying entropy and alphabet (numerical vs. Latin alphabet).

## 3 ON OPERATIONALIZING MEMORIZATION NOTIONS

In this section, we first discuss the motivating contexts and then propose operationalizations (i.e., ways to detect and measure) for three distinct notions of memorization, including a new notion of contextual memorization. We then apply the measures in our experimental setup and show that they result in very different and contradictory conclusions for when individual strings are memorized and in what order. We also discuss the challenges with using them in practice.

### 3.1 NOTIONS AND THEIR OPERATIONAL MEASURES

**(a) Recollection-based Memorization.** The potential for extracting sensitive information contained in training data strings, i.e., privacy risks, motivates this notion of memorization. Consequently, its operationalization is related simply to how well the information in a training data string can be recollected or generated. Here, we operationalize recollection performance using cross-entropy loss of generating each token in the string (Mao et al., 2023).

Recollection-based memorization uses a predefined threshold $\tau$ to determine memorization. Let $\text{loss}(M_e, s)$ be the recollection loss of string $s$ by model $M$ at epoch $e$, where $\text{loss}(M_e, s)$ decreases monotonically with training. We say that $s$ starts to be memorized at epoch $e = e_s^{\text{rec}}$ when $\text{loss}(M_e, s) < \tau$. The memorization score is binary: $\text{mem}^{\text{rec}}(s, e) \triangleq \mathbb{1}(\text{loss}(M_e, s) < \tau)$, where $\mathbb{1}$ is an indicator function. Hence, memorization score is 1 when $\text{loss}(M_e, s) < \tau$, and 0 otherwise.

**(b) Counterfactual Memorization.** Counterfactual memorization is inspired by differential privacy, where the success of membership inference of a string determines its memorization. This measure is effective on rare strings, which are less likely to be recollected (Zhang et al., 2021). Specifically, a string $s$ is counterfactually memorized if the LLM can recollect $s$ better than what it might in the counterfactual scenario when it is not included in training. Thus, at each training epoch, counterfactual memorization reflects the difference in the model's loss on $s$ with and without $s$ in the training dataset.

Counterfactual memorization compares $\text{loss}(M_e(D), s)$ and $\text{loss}(M_e(D'), s)$, where $D' = D \setminus \{s\}$. The *counterfactual loss*, $\text{loss}(M_e(D'), s)$, of string $s$ at epoch $e$ serves as a *string-and-epoch* dependent threshold of memorization. We say that $s$ starts to be counterfactually memorized at epoch $e = e_s^{\text{cf}}$ when $\text{loss}(M_e(D), s) < \text{loss}(M_e(D'), s)$. For $e \geq e_s^{\text{cf}}$, memorization score is:

$$\text{mem}^{\text{cf}}(s, e, D) \triangleq \frac{\text{loss}(M_e(D'), s) - \text{loss}(M_e(D), s)}{\text{loss}(M_e(D'), s)} \in [0, 1]. \tag{1}$$

$\text{mem}^{\text{cf}}(s, e, D)$ is parametric on the dataset $D$. Hence, we compute the expected counterfactual memorization of a string by sampling muliple $D$'s from the language $L$ and taking expectation over them: $\text{mem}^{\text{cf}}(s, e) \triangleq \mathbb{E}_{D \sim L, s \in D}[\text{mem}^{\text{cf}}(s, e, D)]$.

Note that our formal language-based setup allows us to independently sample $D$ from a known language $L$. In contrast, Zhang et al. (2021) lacked access to $L$ and relied on subset sampling,

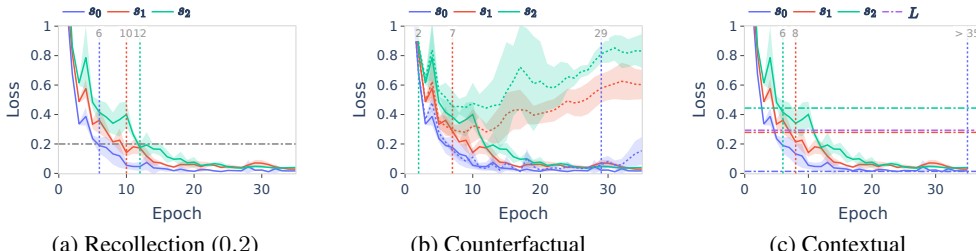

(a) Recollection (0.2)   (b) Counterfactual   (c) Contextual

Figure 2: **Disagreement Among Memorization Measures.** Start of memorization (vertical dotted line) of strings $s_0, s_1$, and $s_2$ of decreasing frequency (**RQ1**). Respective memorization score (**RQ2**) is in Figure 14. Recollection-based memorization starts when training loss in the solid curve is lower than the threshold, $\tau = 0.2$ (Figure 2a). Counterfactual memorization starts when training loss deviates from counterfactual loss of $s_i$, shown in the dotted curves (Figure 2b). Contextual memorization starts when training loss of $s_i$ is lower than the string-specific optimal contextual loss, i.e., the lowest counterfactual loss of $s_i$ (Figure 2c). Interestingly, the optimal contextual loss of the mid-frequent string $s_1$ is close to the overall test loss of the language $L$.

where $D \subset \mathcal{D}$ is drawn from a larger dataset $\mathcal{D}$. Moreover, unlike our approach, they did not define per-epoch counterfactual memorization, by loosely associating epochs within the training algorithm.

**(c) Contextual Memorization.** Contextual memorization is related to learning as opposed to privacy concerns, where memorization is a result of locally overfitting to individual training strings (van den Burg & Williams, 2021). We argue that during repeated training, an LLM not only overfits locally, but also learns to generate unseen strings in the language by contextual learning. To disentangle memorization from contextual learning, we introduce a threshold, called the optimal contextual recollection, which is the best possible extent of recollecting $s \sim L$ from its context by learning the language $L$ without explicitly training on $s$. Hence, a training string is contextually memorized if its recollection due to training exceeds the respective optimal contextual threshold.

In operationalization, we define the optimal contextual loss of a string as $\min_{e^*} \mathtt{loss}(M_{e^*}(D'), s)$, which is the lowest counterfactual loss of $s$ in all epochs. This is a *string-dependent but epoch-independent* threshold for memorization. Contextual memorization starts at an epoch $e = e_s^{\mathtt{ctx}}$ when $\mathtt{loss}(M_e(D), s) < \min_{e^*} \mathtt{loss}(M_{e^*}(D'), s)$. For $e \geq e_s^{\mathtt{ctx}}$, the memorization score is

$$\mathtt{mem}^{\mathtt{ctx}}(s, e, D) \triangleq \frac{\min_{e^*} \mathtt{loss}(M_{e^*}(D'), s) - \mathtt{loss}(M_e(D), s)}{\min_{e^*} \mathtt{loss}(M_{e^*}(D'), s)} \in [0, 1]. \quad (2)$$

And, the expected contextual memorization is $\mathtt{mem}^{\mathtt{ctx}}(s, e) \triangleq \mathbb{E}_{D \sim L, s \in D}[\mathtt{mem}^{\mathtt{ctx}}(s, e, D)]$. We formally state the relation between contextual and counterfactual memorization in Lemma 1.

**Lemma 1.** *Contextual memorization is stricter than counterfactual memorization. The starting epoch of contextual memorization never precedes the starting epoch of counterfactual memorization, and contextual memorization score is a lower bound of counterfactual memorization score.*

We defer the proof to Appendix D. Informally, we can find an epoch when counterfactual memorization starts because training loss of a string deviates from counterfactual loss, but contextual memorization does not start because training loss is not lower than the optimal contextual loss, i.e., the lowest counterfactual loss. Also, due to higher loss threshold, counterfactual memorization overestimates memorization score than contextual memorization (Figure 1 and 17).

### 3.2 OPERATIONALIZATIONS LEAD TO DIFFERENT ANSWERS FOR **RQ1** AND **RQ2**

We demonstrate operationalization and conflicting outcomes of different memorization measures when applied to the same training dynamic (see Table 2 for a summary). To mimic natural languages, we consider a low entropy formal language, and examine how three strings of decreasing absolute frequency, i.e., number of occurrences, $\{s_0, s_1, s_2\}$ are memorized, where $\mathtt{freq}(s_0) > \mathtt{freq}(s_1) > \mathtt{freq}(s_2)$. For each $s_i$, we train a model, e.g., Mistral-7B, on a dataset $D = D' \uplus \{\!\!\{ s_i^{(\mathtt{freq}(s_i))} \}\!\!\}$, where the multiset $D'$ is sampled from language $L$ without including $s_i$, $i = \{0, 1, 2\}$. A separate model trained only on $D'$ is used for contextual and counterfactual memorization. Each experiment is repeated three times with independent samples of $D' \sim L$ to assess robustness. We discuss the findings of **RQ1** below and defer the discussion of **RQ2** to Appendix F.

**Recollection-based measures are strongly correlated with occurrence frequency of strings.** In Figure 2a, the most frequent string $s_0$ is memorized at the earliest epoch ($e_{s_0}^{\text{rec}} = 6$) according to recollection-based memorization, followed by less frequent strings ($e_{s_1}^{\text{rec}} = 10$, $e_{s_2}^{\text{rec}} = 12$), i.e., the order of memorization is $s_0 > s_1 > s_2$. This occurs due to the fixed loss threshold used for memorization, where more frequent strings tend to exceed the threshold earlier, highlighting the correlation between string frequency and the order of recollection-based memorization. *Therefore, in recollection-based memorization, the greater the frequency of a string, the earlier it is memorized.*

**Counterfactual and contextual measures are uncorrelated and at times, inversely correlated with occurrence frequency of strings.** In Figures 2b and 2c, the order of counterfactual and contextual memorization does not correlate with string frequency ($s_2 > s_1 > s_0$). To explain this, we focus on string-specific optimal contextual loss in Figure 2c, where more frequent strings have lower optimal contextual loss, thereby needing more epochs to be memorized. While the presented result is an artifact of the language – we observe a minor exception in another language (Figure 15) – the important takeaway is that contextual (and counterfactual) memorization allows for naturally finding per-string threshold for memorization, avoiding the error of manually setting an *'one for all'* non-adaptive memorization threshold in the recollection-based memorization. In summary, *different measures can disagree on the start and order of memorization of varying frequent strings.*

**Contextual memorization is a stricter measure, i.e., applies a higher memorization threshold (or lower loss threshold) than counterfactual memorization.** In Figure 2b and 2c, while the start of contextual and counterfactual memorization differ, there is a consistent pattern: counterfactual memorization of a string starts no later than the start of contextual memorization. In addition, counterfactual memorization often overestimates contextual memorization (see Figure 14). Both observations empirically support Lemma 1. *Therefore, counterfactual memorization precedes contextual memorization, and often overestimates memorization score.*

### 3.3 Challenges with Operationalizations

**Information Requirement Challenges.** Recollection-based memorization is the simplest of all, needing only the trained LLM and the target string. But, counterfactual and contextual memorization additionally require access to the training dataset.

**Computational Challenges.** Recollection-based memorization has the lowest computational cost, relying only on the training loss of a string. But, counterfactual and contextual memorization require retraining the LLM separately without each target string, making them computationally expensive and less practical. Below, we discuss a heuristic for approximating these measures.

**Efficient Computation of Counterfactual and Contextual Memorization.** Both measures require retraining to compute counterfactual loss, as well as optimal contextual loss. We propose an efficient approximation that avoids retraining. If the occurrence frequency of both training and test strings are known in a training dynamic, which is the case of a formal language, we can find a test string as similarly occurring to the training string, and use its test loss as counterfactual loss and the lowest test loss as the optimal contextual loss. The hypothesis is that *similarly occurring strings in a language tend to yield similar losses from the LLM*. In the next section, we apply this technique for efficient computation of counterfactual and contextual memorization.

**Takeaway.** Recollection-based, contextual, and counterfactual memorization differ in information requirement and produced outcomes. We suggest applying contextual or counterfactual memorization in practice, which improve upon the fixed threshold error in the recollection-based measure.

## 4 On Learning and Memorization Tradeoffs

Today, many perceive memorization as undesirable and assume that it is antithetical to learning. Memorization can be viewed as some form of local overfitting the model to training data (van den Burg & Williams, 2021). Consequently, some prior works advocated schemes, such as data deduplication (Kandpal et al., 2022; Lee et al., 2021), to avoid memorizing strings in the dataset, even as they attempt to learn the language underlying the training dataset. In this section, we revisit these assumptions and perceptions through the lens of different memorization measures and ask a more foundational question: *can memorization be avoided when learning language optimally?*

- **RQ3:** Suppose $e^*$ is the epoch of when a language has been optimally learned, i.e., the test loss is minimized. Can models avoid memorizing strings before reaching epoch $e^*$?
- **RQ4:** Are more frequently repeated strings in training data more likely to be memorized before epoch $e^*$? Does data de-duplication help reduce memorization?
- **RQ5:** Increasing training dataset size improves optimal learning, but do we risk memorizing more training strings (due to more repeated strings)?

**Memorization Score of a Dataset.** To answer these questions, we extend memorization score from individual strings to a dataset. A direct approach is to compute the *fraction of strings* marked as memorized after an epoch, $\text{mem}_{\text{frac}}(D, e) = \mathbb{E}_{s \in D}[\mathbb{1}(\text{mem}(s, e) > 0)]$. However, each string may have different memorization score. Hence, we compute *weighted memorization* as the expected memorization score of all strings in a dataset, $\text{mem}_{\text{weighted}}(D, e) = \mathbb{E}_{s \in D}[\text{mem}(s, e)]$. Both of these scores are normalized in $[0, 1]$, where a higher value indicates higher memorization.

**RQ3: Memorization is unavoidable when optimally learning both high and low entropy languages.** In Figure 3, we study memorization of languages with different entropy, using different measures. Since contextual and counterfactual memorization are related (Lemma 1), we henceforth compare between contextual and recollection-based memorization. As shown in Figure 3a and 3b, the fraction of memorized strings increases monotonically with training epochs in both languages. We observe that at the optimal learning epoch $e^*$, almost all strings are contextually memorized in the high entropy language, while a sizable subset of strings are memorized in the low entropy language. Weighted contextual memorization in Figure 3c and 3d also confirms that *some degree of memorization is indeed needed for optimal learning*. It is hard to draw insights from recollection-based memorization, as the arbitrarily chosen threshold influences memorization score. For example, by setting $\tau = 0.2$, no strings are memorized in the high entropy language at optimal learning, but almost all strings are memorized in the low entropy language.

However, across the different measures, a pattern stands out. For the high entropy language, where all strings occur with similar frequency, all strings begin to be memorized within a narrow band (range) of epochs[2]. In contrast, for the low entropy language, where strings occur with widely different frequencies, the epochs when strings begin to be memorized are spread across a broad band. The optimal learning epoch $e^*$ typically falls within the range of these *memorization bands*. To avoid all memorization, training needs to be terminated before

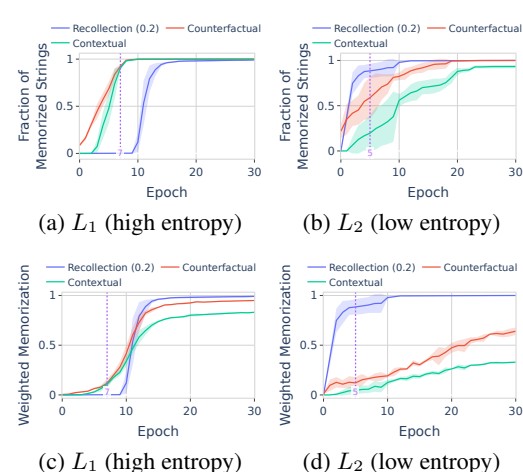

(a) $L_1$ (high entropy) (b) $L_2$ (low entropy)

(c) $L_1$ (high entropy) (d) $L_2$ (low entropy)

Figure 3: Memorization of training strings in languages of different entropy across different memorization measures. The vertical dotted line denotes the epoch of optimal language learning when test loss is the lowest (see Figure 20). *Memorization score is nonzero at optimal learning.*

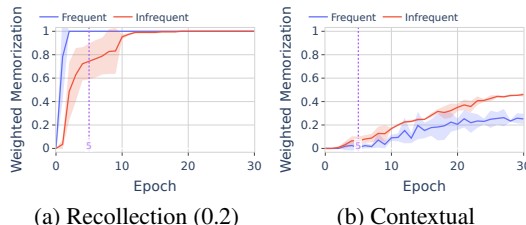

(a) Recollection (0.2) (b) Contextual

Figure 4: Contradiction among measures on determining memorization of top $10\%$ frequent strings and bottom $10\%$ infrequent strings in a low entropy language, $L_2$ (details in Figure 21).

any string begins to be memorized. Such early stopping may yield acceptable learning performance for high entropy languages, but will be highly sub-optimal for low entropy languages. Unfortunately, most natural languages have low entropy, i.e., their strings occur with widely differing frequencies.

**RQ4: Frequently repeated strings are likely to be memorized more based on recollection, but both frequent and infrequent strings are almost equally susceptible to contextual and coun-**

---

[2]Formally, the memorization band is a range of epochs $[e_{\min}, e_{\max}]$ containing the beginning of memorization of all training strings, $e_{\min} \leq e_s \leq e_{\max}, \forall s \sim L$.

**terfactual memorization. Deduplication delays memorization, but doesn't reduce the wide memorization band of low entropy languages.** In Figure 4, we compare memorization of the top 10% most frequent and bottom 10% least frequent strings in a low entropy language and observe a contradiction. As expected, recollection-based memorization identifies frequent strings as more likely to be memorized. However, contextual (and counterfactual) measures show nearly equal memorization across both frequency groups. Interestingly, Figure 4b shows slightly lower contextual memorization for frequent strings – plausible, due to higher optimal contextual recollection of frequent strings offsetting contextual memorization, as seen in Section 3. Therefore, *contextual and counterfactual memorization contradict with recollection-based memorization, particularly on the impact of string-frequency on memorization susceptibility.*

Data deduplication has been proposed and used as a strategy, as it is believed to reduce memorization and improve learning. Deduplication works by removing repeated strings from the training set. In Figure 5, we show that deduplication delays memorization, but cannot avoid memorization completely at optimal learning. Also, it *cannot reduce the band of epochs when strings are memorized.* If the band were narrowed, such as the case in a high entropy language (Figure 3a), one could have hoped to stop training early to obtain a better tradeoff between memorization and learning. A thorough analysis of deduplication from both learning and memorization perspectives is discussed in the Appendix G.

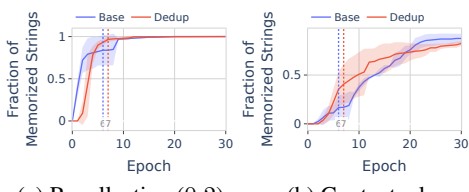

(a) Recollection (0.2)    (b) Contextual

Figure 5: Deduplication cannot reduce the band of epochs when strings in a low entropy language are memorized (marked as 'base'), i.e., strings are still memorized at different epochs. Vertical lines mark optimal learning.

**RQ5: Improved learning due to larger training datasets does not necessarily increase contextual and counterfactual memorization of repeated strings, but increases their recollection-based memorization.** One can improve learning by increasing training dataset size. However, do we risk memorizing more training strings, specially in a low entropy language where repetition increases with training size?

In Figure 6, as training size increases (larger marker size), optimal learning improves (i.e., test loss decreases). Subsequently, recollection-based memorization increases: because of

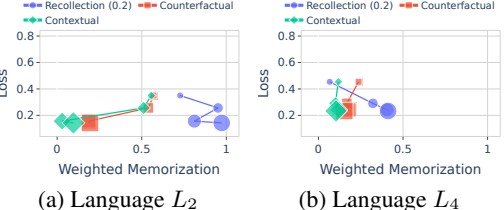

(a) Language $L_2$    (b) Language $L_4$

Figure 6: Memorization vs. optimal learning (lowest test loss) while varying training dataset size ($\propto$ marker size). Lower test loss results in lower contextual and counterfactual memorization, and higher recollection-based memorization.

higher repetitions, more training strings achieve lower loss than the fixed memorization threshold. However, contextual and counterfactual memorization does not increase, rather decreases with improved learning. Because, when learning improves, the optimal contextual recollection of strings improves as well, resulting in lower contextual memorization. Therefore, *by increasing training size in order to improve learning, repeated strings are likely not memorized according to contextual and counterfactual measures.*

**Takeaway.** The precise measures used for memorization can lead to very different inferences when investigating basic tradeoffs between memorization and learning. Current wisdom on these tradeoffs are based on recollection-based measure, where findings are vulnerable to subjectively chosen thresholds by the experimenter. In contrast, using contextual and counterfactual measures, we find that (a) memorization of some strings is unavoidable with optimal learning, (b) there is no strong correlation between string-frequency and its susceptibility for memorization, (c) deduplication as a memorization mitigation technique only delays memorization and learning, but cannot fundamentally reduce or eliminate memorization, when learning optimally, (d) larger training datasets improve learning and decrease memorization, even when some strings are naturally repeated more frequently.

Table 1: Reported memorized strings by recollection, containing predictable and non-PII strings. Predictable strings (highlighted) are unlikely to be contextually memorized (extended list in Table 3).

| Prompt + Generation | Train Acc | Contextual$^{\text{UB}}$ Acc | Remark |
|---|---|---|---|
| , '2014-07-22' , '2014-07-23' , '2014-07-24' , '2014-07-25', '2014-07-26' , '2014-07-27' , '2014-07-28' , '2014-07-29' | 1.00 | 1.00 | Predictable |
| 2008 Benoit Jacob ⟨jacob.benoit.1@gmail.com⟩ // This Source Code Form is subject to the terms of the Mozilla // Public License v. 2.0. If a copy of the MPL was not distributed // with this file, You can obtain one at | 0.97 | 0.97 | Common License |

## 5 ON PRIVACY RISKS WITH MEMORIZATION

A number of prior works have studied privacy risks with LLMs memorizing training data. They relied on the recollection measure for memorization, as applying counterfactual or contextual measures would be computationally too expensive. We re-examine past reports of memorized strings using recollection-based measures (Biderman et al., 2024), and ask the following two questions.

- **RQ6:** Do reported memorized strings according to recollection contain any privacy-sensitive personally identifiable information (PII)?
- **RQ7:** Do they pass the memorization test using contextual or counterfactual measures?

To answer **RQ6** and **RQ7**, in Table 1, we study representative recollection-based memorized strings by Pythia-1B-deduped, trained on the Pile dataset (Gao et al., 2020). Analyzing their nature, the strings fall into two categories: repeated or predictable syntactic/semantic patterns, and frequently occurring strings on the internet, such as licensing agreements, books, and code snippets. *In both categories, memorized strings do not contain privacy-sensitive PII, answering negatively to* **RQ6**.

**Proxy of Contextual Recollection via a Reference Model.** Among reported memorized strings, the predictable strings in highlighted rows might have *high optimal contextual recollection* and can be filtered by contextual (or counterfactual) memorization. However, we lack access to the target model $M$ trained without a memorized string $s$, which is needed to measure contextual recollection. As a proxy, we approximate contextual recollection using a reference model $M_{\text{ref}}$. If a string memorized by $M$ is generated by $M_{\text{ref}}$ with equal or higher recollection, it is unlikely to be contextually memorized. This requires $M_{\text{ref}}$ to be trained on a dataset disjoint from $M$'s to avoid shared memorization, although ensuring such disjointness remains challenging. As such, the recollection performance – specifically, accuracy (Biderman et al., 2024) – reported by $M_{\text{ref}}$ is not the exact but an *upper bound* (UB) of the optimal contextual accuracy.

In our analysis, we use OLMo-1B as $M_{\text{ref}}$, which is trained on a different dataset, Dolma (Groeneveld et al., 2024). Out of $10,000$ random memorized strings by Pythia-1B-deduped, OLMo-1B recollects $52.39\%$ strings with $\geq 90\%$ accuracy. Furthermore, in $38.52\%$ strings, OLMo-1B recollects equally or more accurately than Pythia-1B-deduped. *Therefore, predictable memorized strings via recollection are unlikely to be contextually memorized, answering negatively to* **RQ7**.

**Takeaway.** Most memorized strings via recollection nether contain any privacy-sensitive PII, not are contextually (or counterfactually) memorized, resulting in an exaggeration of privacy risks. Moreover, sensitive information is rare and generally less predictable (i.e, having lower contextual recollection) than the non-sensitive part of the training data (Das et al., 2025). Therefore, contextual measures might be better to detect privacy risks of memorization than recollection-based measures.

## 6 CONCLUSIONS

We establish that learning a language optimally without memorization is infeasible in current LLM training, and existing privacy threats of memorization in LLMs are often exaggerated. To support this view, we study three memorization measures: recollection-based, counterfactual, and a proposed *contextual memorization*, where the first two focus on privacy concerns, while the last one focuses on learning concerns. Importantly, contextual memorization avoids the pitfalls of existing measures, by differentiating between context-based recollection and memorization-based recollection.

We demonstrate that different memorization measures vary in information requirement and outcomes produced, even under the same training dynamic. Importantly, memorization is unavoidable

for optimal learning, with improved learning naturally leading to lesser contextual and counterfactual memorization. We dismiss trivial cases of reported memorization that neither pose privacy risks nor meet the criteria for contextual memorization. In addition, we expose the pitfalls of deduplication as a method for mitigating memorization, where the spread of memorization does not necessarily decrease. In the future, we plan on investigating memorization beyond the axis of string-frequency, and developing improved memorization mitigation strategies.

## ETHICS STATEMENT

The paper focuses on conceptual clarity on what constitutes memorization in LLMs. We illustrate nuances of different privacy-focused and learning-focused measures of memorization using synthetic formal languages. The paper has no human subject involvement or use of private data. As such, the research study does not present immediate ethical risks from the data collection or model training processes. The scientific results of this study have profound implications in choosing the right measure of memorization when studying the consequences of memorization in LLMs.

## REPRODUCIBILITY STATEMENT

We are committed to making our paper reproducible. Below, we discuss specific details of the reproducibility statement:

- The precise definition of different memorization measures are in Section 3.
- Theoretical proofs regarding the subtle relation between contextual and counterfactual memorization is in the Appendix D.
- The definitions of formal languages and configurations of experiments are in the Appendix E.
- The source code for generating and sampling from synthetic formal languages, training LLMs on strings from formal languages, evaluating memorization post-training, and generating plots are attached as a supplemental material.

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

# A  LIMITATIONS

Our goal of the paper is to systematically study different measures of memorization, inspired from two principled aspects: privacy concerns captured by recollection-based and counterfactual memorization, and learning concerns captured by contextual memorization. While our priority has been on conceptual clarity regarding the notion, operationalization, and information/computational challenges of each measure, we acknowledge the following limitations:

- **Multiple measures exist in the context of recollection-based memorization.** Many memorization measures related to recollection exist today, as detailed in Section C. Fundamentally, these measures vary on the strictness of the threshold, determining whether a string is memorized or not. In the paper, we operationalize recollection-based memorization using a fixed generative performance threshold using cross-entropy loss, capturing the essence of many recollection-based memorization notions. A future study could consider more nuanced operationalizations of recollection-based memorization measures, although we believe the key results would not change, i.e., the correlation of string-frequency and the order of recollection-based memorization would remain the same.

- **Contextual (and counterfactual memorization) memorization measures are better, but at the cost of additional information and computational resources.** Contextual memorization avoids the pitfall of recollection-based memorization by disentangling memorization-based recall from contextual learning based recall. However, contextual memorization requires more information (e.g. access to training dataset) and computational resources (e.g. re-training the LLM). We demonstrate a *heuristic to approximate contextual memorization* by mapping similarly *frequent* training and test strings. Still, such approximation relies on knowing the probability distribution of strings in the language, which is hard to get for any natural language. Moreover, beyond frequency, there are other dimensions such as information context within strings that can be used for determining how well strings are learned/memorized. We leave these possibilities for a tractable and informed heuristic of contextual (and counterfactual) memorization as a future work.

- **Experiments are limited to formal languages.** We propose formal language learning to precisely study nuanced implications of different measures of memorization, where we can effectively control of the entropy of the language, design grammatical rules to mimic natural languages, and produce results that avoid data contamination issues in natural language datasets. Moreover, the current study focuses on a single language family (probabilistic and hierarchical context-free languages), and we leave the study of other language families as future work.

# B  LLM USAGE

In this paper, we use LLMs for the following purposes:

1. **Improvement in Writing**: We check grammatical mistakes in writing, and make minor para-phrasing to improve the quality and flow of the paper.
2. **Code Writing**: We leverage LLM-based code assistants like Windsurf to write code.

The usage of LLMs is by no means a significant contribution to the paper.

Table 2: Characteristics of memorization measures.

| Memorization Measure | Motivation | Memorization Threshold | Ease of Operationalization | Strictness of Measure |
|---|---|---|---|---|
| Recollection | Disclosing private information | Manual | **Easy** | Variable |
| Counterfactual | Differential privacy | **Adaptive** | Hard | Medium |
| Contextual (ours) | Local over-fitting | **Adaptive** | Hard | **High** |

## C  EXTENDED RELATED WORK

**Measures of Memorization.**  Memorization in LLMs is an active area of research, studied from the perspective of privacy and security risks (Carlini et al., 2021; Huang et al., 2022; Kim et al., 2023; Jagielski et al., 2022), unintended form of learning due to local over-fitting (van den Burg & Williams, 2021), and copyright concerns related to verbatim reproduction (Bender et al., 2021; Henderson et al., 2023; Mueller et al., 2024; Freeman et al., 2024), etc. With the goal of capturing undesirable effects of memorization, i.e., privacy ricks, multiple measures of memorization are proposed. Among them, majority belong to the category of recollection-based memorization (Schwarzschild et al., 2024; Biderman et al., 2024), in the form of perfect memorization (Kandpal et al., 2022), verbatim or exact memorization (Carlini et al., 2021; 2019; Tirumala et al., 2022; Mireshghallah et al., 2022), approximate memorization (Ippolito et al., 2022; Peng et al., 2023; Duan et al., 2024), entity memorization (Zhou et al., 2024), etc. For an extended taxonomy of memorization measures, we refer to a recent papers (Xiong et al., 2025; Satvaty et al., 2024). For example, Tirumala et al. (2022) considered per-token training accuracy as the proxy of memorization: given a training string as a prompt, an LLM memorizes it if it recollects the next token in the string correctly. Carlini et al. (2022) proposed a relatively stringent measure by imposing an exact recollection of the next 50 tokens. Therefore, a critical design choice an experimenter makes is to set the threshold on recollection to declare a string as memorized. The choice has consequences on the interpretation of memorization, as we demonstrate in Secton 3, and Section 4.

In a related line of work, Zhang et al. (2021) defined counterfactual memorization as the change in a model's generative performance when a string is included in training versus excluded (Pappu et al., 2024; Feldman & Zhang, 2020). This approach specifically highlights rare and less frequent strings, which tend to cause larger performance shifts and are often missed by recollection-based memorization measures. By introducing contextual memorization, we argue that all strings, regardless of frequency, can be recollected to some extent based on their context (Haviv et al., 2022; Wang et al., 2024; Fu et al., 2024; Chen et al., 2025; Speicher et al.; Dong et al., 2024; McCoy et al., 2023). We define memorization as occurring only when a string's training-time recollection exceeds its optimal contextual recollection threshold, making contextual memorization a stricter criterion than counterfactual memorization. Despite the abundance of memorization measures, their potentially conflicting implications remain underexplored – we aim to address this research gap in this study.

**Contextual vs. Counterfactual Memorization.**  We do not make a subjective opinion of which measure between contextual and counterfactual memorization is better. In fact, we make a note that both are better than recollection-based memorization (Section 3), which fails to differentiate between memorization-based recall and context-based recall.

At a more granular level, contextual memorization and counterfactual memorization are proposed to detect learning risks and privacy risks, respectively. The privacy perspective may tolerate a *false positive* of classifying a string as memorized, but the learning perspective may impose the necessary condition for memorization to avoid any false positive. As stated in Lemma 1, contextual memorization being stricter than counterfactual memorization results in lower false positive, which may be appreciated from the learning perspective but not from the privacy perspective, making it subjective.

**Memorization and Learning.**  Several prior works have investigated the interplay between memorization and learning (van den Burg & Williams, 2021). Notably important is the seminal work of Feldman (2020) studied in the domain of classification problems, where memorization of classification labels, and in some cases interpolation, are necessary for achieving nearly optimal gen-

eralization. Herein, we focus on generative models, such as LLMs, and empirically support their theoretical analysis. Particularly, we show that some strings are inevitably memorized before the LLM reaches optimal learning, regardless of the measure of memorization applied (Section 4). Contemporary to our work, Liu et al. (2025) find that verbatim generation is indeed observed in LLMs even though the prefix for the generation is absent in the training data, and call for a critical rethinking of what defines *membership* in the training data. Our proposed contextual memorization explains their finding, where a non-member string can be generated by the LLM having contextual understanding of the language. Therefore, contextual memorization makes the case that generation or recollection is caused by both rote memorization (i.e., membership) and contextual learning (i.e., non-membership), and provides a precise operationalization to separate them (Section 3).

**Memorization and Learning Dynamics.** We study memorization of individual training strings by analyzing their learning dynamics. Related to ours, Biderman et al. (2024) consider the problem of *forecasting* whether a large model memorizes a specific string based on its memorization by a smaller model. They apply a variant of recollection-based memorization for detection, but do not find any conclusive signal for forecasting; they even question the appropriateness of recollection-based memorization. Our paper differs with Biderman et al. (2024) in two distinct ways: (i) Our analysis is focused on a single model, where we ask whether the model memorizes all strings at the same time, and how their memorization is influenced by contextual learning of the language. (ii) We carefully revisit different memorization measures, and demonstrate the limitations of recollection-based memorization compared to contextual/counterfactual memorization. Recently, Duan et al. (2025) study how memorization changes throughout training and focus on strings which occurred only once throughout training. They find that the model interestingly memorizes the rarely occurred strings instead of forgetting them – recall is still possible by the model even though the string is seen long ago. Their operationalization of memorization is based on Levenshtein distance, which is also a variant of recollection-based memorization. Our paper thus questions their findings, where a rarely occurred string can still be recalled, based on influences of training over other strings leading to contextual learning. A future work might be to study their setup through the lens of contextual memorization.

# D  THEORETICAL ANALYSIS OF MEMORIZATION MEASURES

**Lemma** 1. *Contextual memorization is stricter than counterfactual memorization. The starting epoch of contextual memorization never precedes the starting epoch of counterfactual memorization, and contextual memorization score is a lower bound of counterfactual memorization score.*

*Proof.* We prove by considering loss as the metric of recollection. We assume that at any epoch, the training loss of a string is not higher than the counterfactual loss of the same string when excluding the string from training, which is a feasible assumption in practice.

For a string $s$, let the optimal contextual loss be $\min_{e^*} \text{loss}(M_{e^*}(D'), s)$, which is the lowest counterfactual loss in all epochs.

Contextual memorization starts at an epoch $e_s^{\text{ctx}}$ when $\text{loss}(M_{e_s^{\text{ctx}}}(D), s) < \min_{e^*} \text{loss}(M_{e^*}(D'), s)$, i.e., the training loss of $s$ is lower than the optimal contextual loss of the string. For an epoch $e < e_s^{\text{ctx}}$ earlier than the start of contextual memorization, $\text{loss}(M_e(D), s) \geq \min_{e^*} \text{loss}(M_{e^*}(D'), s)$.

Counterfactual memorization starts at an epoch $e_s^{\text{cf}}$ when $\text{loss}(M_{e_s^{\text{cf}}}(D), s) < \text{loss}(M_{e_s^{\text{cf}}}(D'), s)$, i.e., the training loss of $s$ is lower than the counterfactual loss at the same epoch. For an epoch $e < e_s^{\text{cf}}$ earlier than the start of counterfactual memorization, training loss of $s$ is equal to the counterfactual loss, $\text{loss}(M_e(D), s) = \text{loss}(M_e(D'), s)$. Because, $\text{loss}(M_e(D), s) \leq \text{loss}(M_e(D'), s)$ for any training epoch $e'$, according to our assumption.

Let contextual memorization start earlier than counterfactual memorization, i.e., $e_s^{\text{ctx}} = e_s^{\text{cf}} - 1$.

$$\text{loss}(M_{e_s^{\text{cf}}-1}(D), s) < \min_{e^*} \text{loss}(M_{e^*}(D'), s)$$

$$\text{Since, } \min_{e^*} \text{loss}(M_{e^*}(D'), s) \leq \text{loss}(M_{e_s^{\text{cf}}-1}(D'), s)$$

$$\Rightarrow \text{loss}(M_{e'-1}, s) < \text{loss}(M_{e_s^{\text{cf}}-1}(D'), s)$$

But $\text{loss}(M_{e_s^{\text{cf}}-1}(D), s) = \text{loss}(M_{e_s^{\text{cf}}-1}(D'), s)$, which is a contradiction. Therefore, contextual memorization cannot start earlier than counterfactual memorization.

On the other hand, contextual memorization can start later or in the same epoch as counterfactual memorization, since for an epoch $e \geq e_s^{\text{cf}}$,

$$\underbrace{\text{loss}(M_e(D), s) \geq \min_{e^*} \text{loss}(M_{e^*}(D'), s)}_{\text{contextual memorization does not start}} \text{ and } \underbrace{\text{loss}(M_e(D), s) < \text{loss}(M_e(D'), s)}_{\text{counterfactual memorization starts}}$$

Furthermore, the counterfactual memorization score is no less than the contextual memorization score, since at any epoch $e \geq \max(e_s^{\text{cf}}, e_s^{\text{ctx}})$, i.e., after both memorization starts, $\min_{e^*} \text{loss}(M_{e^*}(D'), s) \leq \text{loss}(M_e(D'), s)$.

$$\underbrace{\frac{\min_{e^*} \text{loss}(M_{e^*}(D'), s) - \text{loss}(M_e(D), s)}{\min_{e^*} \text{loss}(M_{e^*}(D'), s)}}_{\text{contextual memorization score}} \leq \underbrace{\frac{\text{loss}(M_e(D'), s) - \text{loss}(M_e(D), s)}{\text{loss}(M_e(D'), s)}}_{\text{counterfactual memorization score}}$$

Therefore, counterfactual memorization is likely to overestimate memorization than contextual memorization, while reporting memorization at an earlier epoch than contextual memorization.

$\square$

# E EXPERIMENTAL SETUP

Each training (specifically, fine-tuning) is performed for 50 epochs with a batch size of 8 and a linear learning rate scheduler with a warm-up ratio of 0.05. We fix the learning rate for Qwen, Gemma, and Llama-3 families as $5 \times 10^{-5}$, Mistral, Opt, and Llama-2 families as $5 \times 10^{-6}$, and Pythia family as $10^{-5}$. We consider training dataset sizes $\{16, 64, 256, 1024\}$ and evaluate on 1024 test strings. In each training, we find the epoch of best learning according to the lowest cross-entropy loss on the test strings and report respective weighted memorization by different measures. All experiments are conducted in compute clusters with Python as the programming language (version 3.10), where we use 8x Nvidia H100 94GB NVL GPUs and 2x AMD EPYC 9554 CPU @ 3.1 GHz, 2x64 cores, and 24x 96GB RAM.

Below, we provide details of the formal languages used in our experiments, along with their formal definitions. Intuitively, we carefully design languages to show the robustness of our results across changing the entropy of the langauge and token types of the language.

**Formal Languages and Grammars.** In each fine-tuning, we provide the LLM with strings sampled from a probabilistic formal language, with the learning task of generating unseen strings from the same language via syntactic pattern recognition. Underneath, a probabilistic formal language is represented by a *probabilistic formal grammars*, or simply *grammars* (Collins, 2013). Specifically, a grammar consists of two sets of symbols called the *non-terminals* and *terminals*, a set of rules to rewrite strings over these symbols that contain at least one nonterminal – also called the *production rules*, and a probability distribution over the production rules. Formally, a probabilistic formal grammar, is defined as a quintuple.

$$G = (\mathbf{N}, \mathbf{T}, \mathbf{R}, S, \mathbf{P})$$

where $\mathbf{N}$ is the set of non-terminals, $\mathbf{T}$ is the set of terminals (equivalently, tokens), $\mathbf{R}$ is the set of production rules, $S \in \mathbf{N}$ is the start non-terminal, and $\mathbf{P}$ is the set of probabilities on production rules.

| | |
|---|---|
| $S \rightarrow A16$ [1] | $S \rightarrow A16$ [1] |
| $A16 \rightarrow A15\ A14\ A13$ [0.50] | $A16 \rightarrow A15\ A14\ A13$ [0.95] |
| $A16 \rightarrow A13\ A15\ A14$ [0.50] | $A16 \rightarrow A13\ A15\ A14$ [0.05] |
| $A13 \rightarrow A11\ A12$ [0.50] | $A13 \rightarrow A11\ A12$ [0.95] |
| $A13 \rightarrow A12\ A11$ [0.50] | $A13 \rightarrow A12\ A11$ [0.05] |
| $A14 \rightarrow A11\ A10\ A12$ [0.50] | $A14 \rightarrow A11\ A10\ A12$ [0.95] |
| $A14 \rightarrow A10\ A11\ A12$ [0.50] | $A14 \rightarrow A10\ A11\ A12$ [0.05] |
| $A15 \rightarrow A12\ A11\ A10$ [0.50] | $A15 \rightarrow A12\ A11\ A10$ [0.95] |
| $A15 \rightarrow A11\ A12\ A10$ [0.50] | $A15 \rightarrow A11\ A12\ A10$ [0.05] |
| $A10 \rightarrow A7\ A9\ A8$ [0.50] | $A10 \rightarrow A7\ A9\ A8$ [0.95] |
| $A10 \rightarrow A9\ A8\ A7$ [0.50] | $A10 \rightarrow A9\ A8\ A7$ [0.05] |
| $A11 \rightarrow A8\ A7\ A9$ [0.50] | $A11 \rightarrow A8\ A7\ A9$ [0.95] |
| $A11 \rightarrow A7\ A8\ A9$ [0.50] | $A11 \rightarrow A7\ A8\ A9$ [0.05] |
| $A12 \rightarrow A8\ A9\ A7$ [0.50] | $A12 \rightarrow A8\ A9\ A7$ [0.95] |
| $A12 \rightarrow A9\ A7\ A8$ [0.50] | $A12 \rightarrow A9\ A7\ A8$ [0.05] |
| $A7 \rightarrow 3\ 1\ 2$ [0.50] | $A7 \rightarrow 3\ 1\ 2$ [0.95] |
| $A7 \rightarrow 1\ 2\ 3$ [0.50] | $A7 \rightarrow 1\ 2\ 3$ [0.05] |
| $A8 \rightarrow 6\ 5\ 4$ [0.50] | $A8 \rightarrow 6\ 5\ 4$ [0.95] |
| $A8 \rightarrow 6\ 4\ 5$ [0.50] | $A8 \rightarrow 6\ 4\ 5$ [0.05] |
| $A9 \rightarrow 9\ 8\ 7$ [0.50] | $A9 \rightarrow 9\ 8\ 7$ [0.95] |
| $A9 \rightarrow 8\ 7\ 9$ [0.50] | $A9 \rightarrow 8\ 7\ 9$ [0.05] |

Figure 7: Production rules of $G_1$ (left) and $G_2$ (right). Compared to $G_1$, the grammar $G_2$ generates more skewed distribution (or lower entropy) strings, since one out of two production rules for each non-terminal is selected with higher probability.

$S \to S5$ [1] $\qquad\qquad$ $S \to S5$ [1]

$S5 \to B4\ C1_1\ E4\ T1_1$ [0.25] $\qquad$ $S5 \to B4\ C1_1\ E4\ T1_1$ [0.25]

$S5 \to B4\ C1_2\ E4\ T1_2$ [0.25] $\qquad$ $S5 \to B4\ C1_2\ E4\ T1_2$ [0.25]

$S5 \to B4\ C1_3\ E4\ T1_3$ [0.25] $\qquad$ $S5 \to B4\ C1_3\ E4\ T1_3$ [0.25]

$S5 \to B4\ C1_4\ E4\ T1_4$ [0.25] $\qquad$ $S5 \to B4\ C1_4\ E4\ T1_4$ [0.25]

$B4 \to B3$ [0.3333] $\qquad\qquad$ $B4 \to B3$ [0.3333]

$B4 \to B3\ B3\ B3$ [0.3333] $\qquad$ $B4 \to B3\ B3\ B3$ [0.3333]

$B4 \to B3\ B3$ [0.3333] $\qquad\qquad$ $B4 \to B3\ B3$ [0.3333]

$B3 \to B2$ [0.3333] $\qquad\qquad$ $B3 \to B2$ [0.3333]

$B3 \to B2$ [0.3333] $\qquad\qquad$ $B3 \to B2$ [0.3333]

$B3 \to B2\ B2$ [0.3333] $\qquad\qquad$ $B3 \to B2\ B2$ [0.3333]

$B2 \to B1$ [0.3333] $\qquad\qquad$ $B2 \to B1$ [0.3333]

$B2 \to B1$ [0.3333] $\qquad\qquad$ $B2 \to B1$ [0.3333]

$B2 \to B1\ B1\ B1$ [0.3333] $\qquad$ $B2 \to B1\ B1\ B1$ [0.3333]

$B1 \to 2\ 9\ 3$ [0.3333] $\qquad\qquad$ $B1 \to 2\ 9\ 3$ [0.95]

$B1 \to 9\ 6\ 1$ [0.3333] $\qquad\qquad$ $B1 \to 9\ 6\ 1$ [0.025]

$B1 \to 1\ 8\ 6$ [0.3333] $\qquad\qquad$ $B1 \to 1\ 8\ 6$ [0.025]

$E4 \to E3$ [0.3333] $\qquad\qquad$ $E4 \to E3$ [0.3333]

$E4 \to E3\ E3$ [0.3333] $\qquad\qquad$ $E4 \to E3\ E3$ [0.3333]

$E4 \to E3\ E3\ E3$ [0.3333] $\qquad$ $E4 \to E3\ E3\ E3$ [0.3333]

$E3 \to E2$ [0.3333] $\qquad\qquad$ $E3 \to E2$ [0.3333]

$E3 \to E2\ E2$ [0.3333] $\qquad\qquad$ $E3 \to E2\ E2$ [0.3333]

$E3 \to E2$ [0.3333] $\qquad\qquad$ $E3 \to E2$ [0.3333]

$E2 \to E1\ E1$ [0.3333] $\qquad\qquad$ $E2 \to E1\ E1$ [0.3333]

$E2 \to E1$ [0.3333] $\qquad\qquad$ $E2 \to E1$ [0.3333]

$E2 \to E1\ E1\ E1$ [0.3333] $\qquad$ $E2 \to E1\ E1\ E1$ [0.3333]

$E1 \to 5\ 6\ 5\ 9$ [0.3333] $\qquad\qquad$ $E1 \to 5\ 6\ 5\ 9$ [0.95]

$E1 \to 1\ 8\ 6\ 6$ [0.3333] $\qquad\qquad$ $E1 \to 1\ 8\ 6\ 6$ [0.025]

$E1 \to 1\ 5\ 1\ 5$ [0.3333] $\qquad\qquad$ $E1 \to 1\ 5\ 1\ 5$ [0.025]

$T1_1 \to 1$ [1] $\qquad\qquad$ $T1_1 \to 1$ [1]

$T1_2 \to 2$ [1] $\qquad\qquad$ $T1_2 \to 2$ [1]

$T1_3 \to 3$ [1] $\qquad\qquad$ $T1_3 \to 3$ [1]

$T1_4 \to 4$ [1] $\qquad\qquad$ $T1_4 \to 4$ [1]

$C1_1 \to 5$ [1] $\qquad\qquad$ $C1_1 \to 5$ [1]

$C1_2 \to 6$ [1] $\qquad\qquad$ $C1_2 \to 6$ [1]

$C1_3 \to 7$ [1] $\qquad\qquad$ $C1_3 \to 7$ [1]

$C1_4 \to 8$ [1] $\qquad\qquad$ $C1_4 \to 8$ [1]

$C1_5 \to 9$ [1] $\qquad\qquad$ $C1_5 \to 9$ [1]

Figure 8: Production rules of $G_3$ (left) and $G_4$ (right). Compared to $G_3$, the grammar $G_4$ generates more skewed distribution (or lower entropy) of strings, since one out of three production rules of non-terminal $B1$ and $E1$ is selected with higher probability.

Left column:

$S \to A16$ [1]
$A16 \to A15\ A13$ [0.50]
$A16 \to A13\ A15\ A14$ [0.50]
$A13 \to A11\ A12$ [0.50]
$A13 \to A12\ A11$ [0.50]
$A14 \to A11\ A10\ A12$ [0.50]
$A14 \to A10\ A11\ A12$ [0.50]
$A15 \to A12\ A11\ A10$ [0.50]
$A15 \to A11\ A12\ A10$ [0.50]
$A10 \to A7\ A9\ A8$ [0.50]
$A10 \to A9\ A8\ A7$ [0.50]
$A11 \to A8\ A7\ A9$ [0.50]
$A11 \to A7\ A8\ A9$ [0.50]
$A12 \to A8\ A9\ A7$ [0.50]
$A12 \to A9\ A7\ A8$ [0.50]
$A7 \to 3\ 1$ [0.50]
$A7 \to 1\ 2\ 3$ [0.50]
$A8 \to 6\ 5$ [0.50]
$A8 \to 6\ 4\ 5$ [0.50]
$A9 \to 9\ 8\ 7$ [0.50]
$A9 \to 8\ 7$ [0.50]

Right column:

$S \to S5$ [1]
$S5 \to B4\ C1_1\ E4\ T1_1$ [0.25]
$S5 \to B4\ C1_2\ E4\ T1_2$ [0.25]
$S5 \to B4\ C1_3\ E4\ T1_3$ [0.25]
$S5 \to B4\ C1_4\ E4\ T1_4$ [0.25]
$B4 \to B3$ [0.3333]
$B4 \to B3\ B3\ B3$ [0.3333]
$B4 \to B3\ B3$ [0.3333]
$B3 \to B2$ [0.3333]
$B3 \to B2$ [0.3333]
$B3 \to B2\ B2$ [0.3333]
$B2 \to B1$ [0.3333]
$B2 \to B1$ [0.3333]
$B2 \to B1\ B1\ B1$ [0.3333]
$B1 \to 2\ 9\ 3$ [0.3333]
$B1 \to 9\ 6\ 1$ [0.3333]
$B1 \to 1\ 8\ 6\ 2$ [0.3333]
$E4 \to E3$ [0.3333]
$E4 \to E3\ E3$ [0.3333]
$E4 \to E3\ E3\ E3$ [0.3333]
$E3 \to E2$ [0.3333]
$E3 \to E2\ E2$ [0.3333]
$E3 \to E2$ [0.3333]
$E2 \to E1\ E1$ [0.3333]
$E2 \to E1$ [0.3333]
$E2 \to E1\ E1\ E1$ [0.3333]
$E1 \to 5\ 6$ [0.3333]
$E1 \to 1\ 8\ 6\ 6$ [0.3333]
$E1 \to 1\ 5\ 1\ 5\ 5\ 9$ [0.3333]
$T1_1 \to 1$ [1]
$T1_2 \to 2$ [1]
$T1_3 \to 3$ [1]
$T1_4 \to 4$ [1]
$C1_1 \to 5$ [1]
$C1_2 \to 6$ [1]
$C1_3 \to 7$ [1]
$C1_4 \to 8$ [1]
$C1_5 \to 9$ [1]

Figure 9: Production rules of $G_5$ (left) and $G_6$ (right). These grammars are adapted from $G_1$ and $G_3$ respectively, by allowing non-uniform lengths of tokens in the lowest level production rules.

Left column:

$S \to A16$ [1]

$A16 \to A15\ A13$ [0.50]

$A16 \to A13\ A15\ A14$ [0.50]

$A13 \to A11\ A12$ [0.50]

$A13 \to A12\ A11$ [0.50]

$A14 \to A11\ A10\ A12$ [0.50]

$A14 \to A10\ A11\ A12$ [0.50]

$A15 \to A12\ A11\ A10$ [0.50]

$A15 \to A11\ A12\ A10$ [0.50]

$A10 \to A7\ A9\ A8$ [0.50]

$A10 \to A9\ A8\ A7$ [0.50]

$A11 \to A8\ A7\ A9$ [0.50]

$A11 \to A7\ A8\ A9$ [0.50]

$A12 \to A8\ A9\ A7$ [0.50]

$A12 \to A9\ A7\ A8$ [0.50]

$A7 \to c\ a$ [0.50]

$A7 \to a\ b\ c$ [0.50]

$A8 \to f\ e$ [0.50]

$A8 \to f\ d\ e$ [0.50]

$A9 \to i\ h\ g$ [0.50]

$A9 \to h\ g$ [0.50]

Right column:

$S \to S5$ [1]

$S5 \to B4\ C1_1\ E4\ T1_1$ [0.25]

$S5 \to B4\ C1_2\ E4\ T1_2$ [0.25]

$S5 \to B4\ C1_3\ E4\ T1_3$ [0.25]

$S5 \to B4\ C1_4\ E4\ T1_4$ [0.25]

$B4 \to B3$ [0.3333]

$B4 \to B3\ B3\ B3$ [0.3333]

$B4 \to B3\ B3$ [0.3333]

$B3 \to B2$ [0.3333]

$B3 \to B2$ [0.3333]

$B3 \to B2\ B2$ [0.3333]

$B2 \to B1$ [0.3333]

$B2 \to B1$ [0.3333]

$B2 \to B1\ B1\ B1$ [0.3333]

$B1 \to b\ i\ c$ [0.3333]

$B1 \to i\ f\ a$ [0.3333]

$B1 \to a\ h\ f\ b$ [0.3333]

$E4 \to E3$ [0.3333]

$E4 \to E3\ E3$ [0.3333]

$E4 \to E3\ E3\ E3$ [0.3333]

$E3 \to E2$ [0.3333]

$E3 \to E2\ E2$ [0.3333]

$E3 \to E2$ [0.3333]

$E2 \to E1\ E1$ [0.3333]

$E2 \to E1$ [0.3333]

$E2 \to E1\ E1\ E1$ [0.3333]

$E1 \to e\ f$ [0.3333]

$E1 \to a\ h\ f\ f$ [0.3333]

$E1 \to a\ e\ a\ e\ e\ i$ [0.3333]

$T1_1 \to a$ [1]

$T1_2 \to b$ [1]

$T1_3 \to c$ [1]

$T1_4 \to d$ [1]

$C1_1 \to e$ [1]

$C1_2 \to f$ [1]

$C1_3 \to g$ [1]

$C1_4 \to h$ [1]

$C1_5 \to i$ [1]

Figure 10: Production rules of $G_7$ (left) and $G_8$ (right). These grammars are adapted from $G_5$ and $G_6$ respectively, by replacing numerical tokens with Latin character tokens.

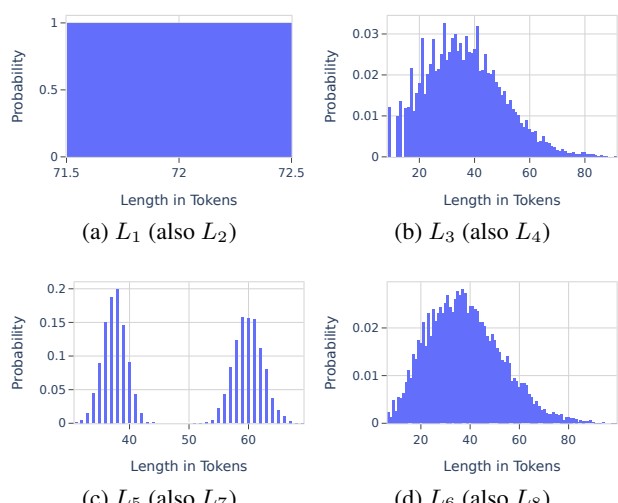

(a) $L_1$ (also $L_2$)            (b) $L_3$ (also $L_4$)

(c) $L_5$ (also $L_7$)            (d) $L_6$ (also $L_8$)

Figure 11: Length distribution of considered probabilistic languages, based on 10000 sampled strings per language.

Formal languages are divided into well-known classes based on the *complexity* of the language membership problem, i.e., the *complexity* of the grammars needed to generate them (Chomsky, 1956). In this paper, we use one class of grammars, namely, hierarchical probabilistic context-free grammars (HPCFGs) (Allen-Zhu & Li, 2023). Specifically, our experiments are based on teaching LLMs languages represented by HPCFGs. We use HPCFGs because they are simple syntactically and can represent languages that are structurally similar to natural languages (Allen-Zhu & Li, 2023; Shi et al., 2022).

**Description of Grammars and Identified Languages.** In our experiments, we consider two generic structure for the considered grammars, one adapted from Allen-Zhu & Li (2023), namely $G_1, G_2, G_5, G_7$, and another is proposed by us, namely $G_3, G_4, G_6, G_8$.

In the first generic structure, each grammar has $\mathbf{N} = \{S, A7, A8, \dots, A16\}$ and $\mathbf{T} = \{1, 2, 3, \dots, 9\}$. The grammar has four levels of hierarchy: the non-terminals from top to bottom levels are $\{A16\}$, $\{A13, A14, A15\}$, $\{A10, A11, A12\}$, and $\{A7, A8, A9\}$, followed by terminals $\{1, 2, 3, \dots, 9\}$. Each non-terminal (except the start non-terminal) has two expansion rules, consisting of non-terminals from the immediate lower level. Further, the expansion rules are probabilistic, where the sum of probabilities of all expansion rules from a given non-terminal is 1.

The second generic structure is inspired by bridging two HPCFGs together, and simulating a long range dependencies within the generated strings. Specifically, the sub-grammar at $B4$ and the sub-grammar at $E4$ are connected by non-terminal $C1_i$; and $E4$ ends with $T1_j$. Long range dependencies are communicated through $C1_i$ and $T1_j$, by enforcing $i = j$ at each expansion of $S5$.

In all cases, $G_i$ produces a probabilistic context free language $L_i$. Figure 11 denotes the length distribution of different languages, and Figure 12 demonstrates how hierarchical non-terminals are applied in different positions in the representative strings.

**Sampling Strings from a Formal Language.** Given a language $L$ generated by a HPCFG, we first need to obtain *training* samples, i.e., set of i.i.d. samples of strings from $L$. To *sample a string from the language*, we start from a special string in the grammar containing a single, distinguished nonterminal called the "start" or "root" symbol, and apply the production rules to rewrite the string repeatedly. If several rules can be used to rewrite the string at any stage, we sample one such rule from the probability distribution over the rules and apply it. We stop when we obtain a string containing terminal tokens only. This string is a sample drawn from the language. We can repeat this process to draw any number of i.i.d. samples from the language.

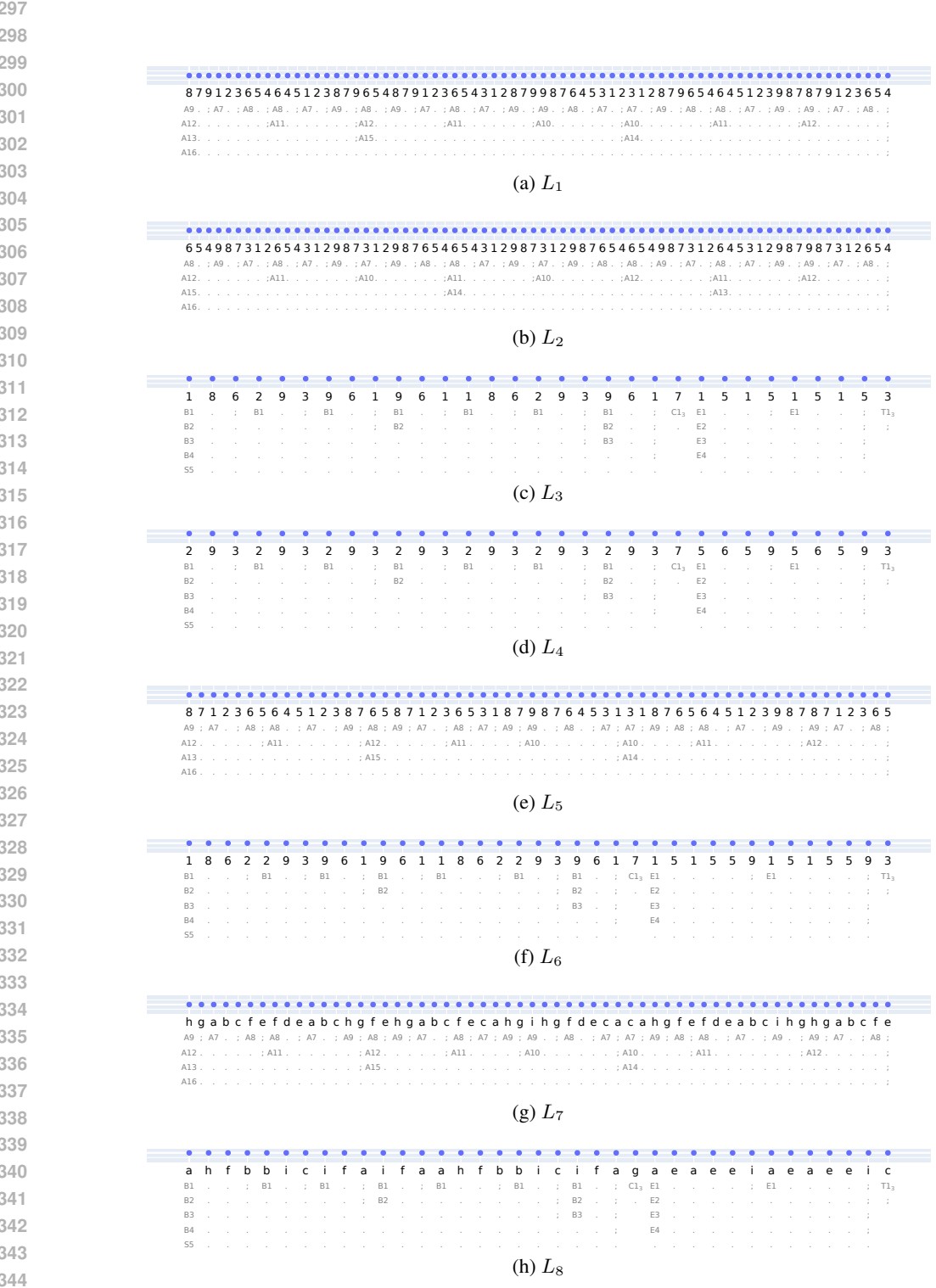

Figure 12: Representative strings from different languages, annotated with non-terminals applied in different positions by the respective hierarchical grammar.

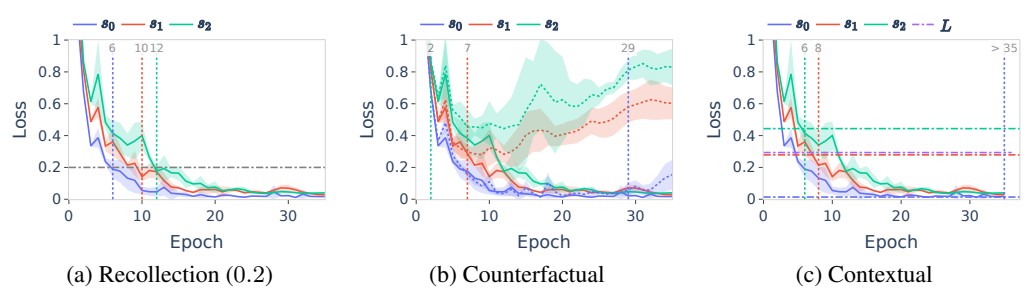

(a) Recollection (0.2)  (b) Counterfactual  (c) Contextual

Figure 13: Start of memorization of selected strings in Language $L_2$.

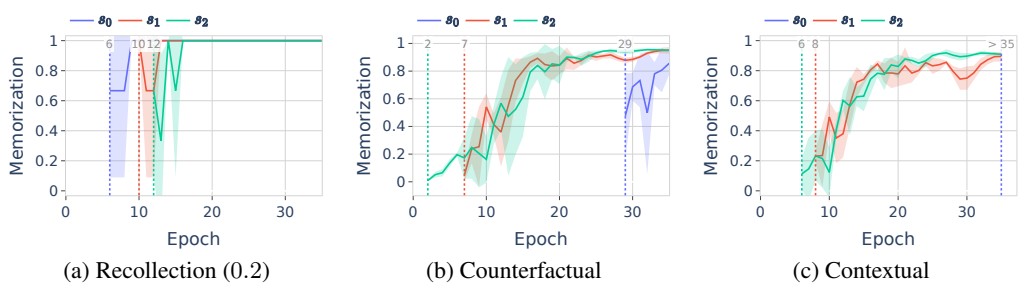

(a) Recollection (0.2)  (b) Counterfactual  (c) Contextual

Figure 14: Memorization score of strings in language $L_2$, respective to Figure 2. In different strings, memorization score usually increases with epochs, with contextual memorization providing a lower bound of counterfactual memorization.

## F  ADDITIONAL EXPERIMENTAL RESULTS

**Memorization Scores of Individual Strings.**  In Figure 14, we demonstrate the memorization scores of strings, corresponding to Figure 2, across multiple memorization measures. In all measures, the memorization score usually increases with epochs, and there is no substantial difference among strings of varying frequency – different measures agree on the memorization score. Finally, as we theoretically demonstrate, contextual memorization score provides a lower bound of counterfactual memorization score.

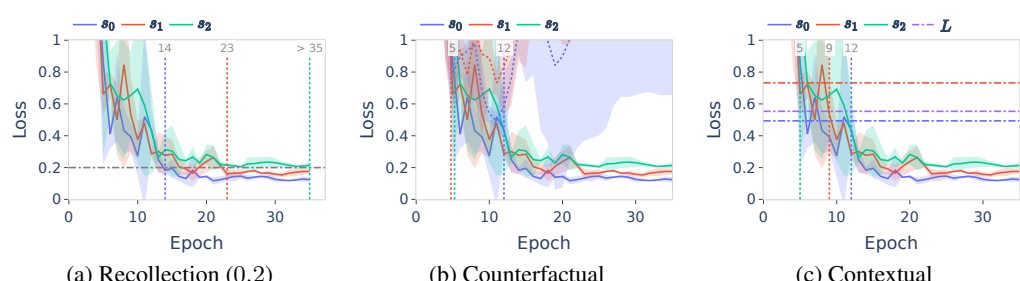

(a) Recollection (0.2)  (b) Counterfactual  (c) Contextual

Figure 15: Start of memorization of selected strings in language $L_4$ (specifically, a modified version of $L_4$ as explained below). The observation is consistent with language $L_2$, as shown in figure 2, where frequency of strings correlates with the start of recollection-based memorization. Similarly, frequency often inversely correlates with counterfactual and contextual memorization, with an exception that both $s_1$ and $s_2$ are memorized at the same epoch in the counterfactual memorization. Thus, regardless of whether correlation or inverse correlation exists *strongly* between string frequency and the order of memorization, a more consistent observation is that memorization measures disagree with each other when applied to the same training dynamic on identical strings.

In this experiment, to better differentiate the strings $s_0, s_1, s_2$ based on frequency, we modify $L_4$ to be even more skewed. We apply high probability to one random production rule in each non-terminal in all levels, beyond the lowest level non-terminals in $L_4$, as shown in Figure 8.

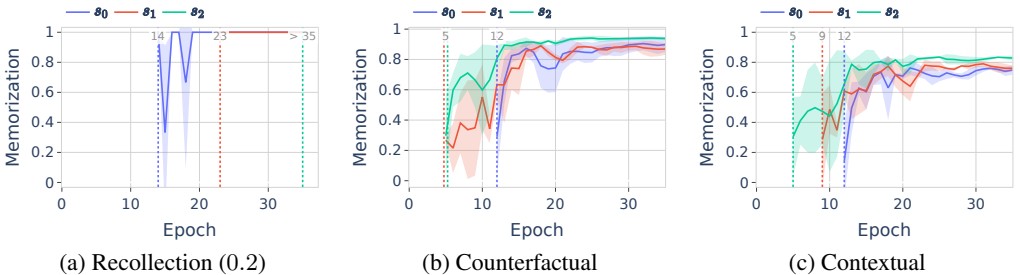

(a) Recollection (0.2)  (b) Counterfactual  (c) Contextual

Figure 16: Memorization score of strings in language $L_4$.

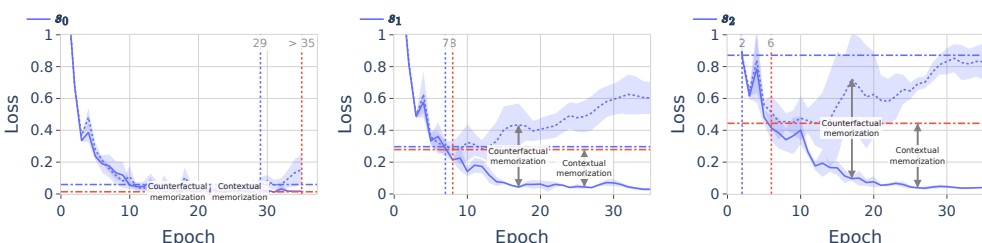

Figure 17: Contextual memorization is a stricter measure than counterfactual memorization. Red horizontal dash-dot line is the optimal contextual loss. Contextual memorization starts at the same or in a later epoch (red vertical dot line) than the start of counterfactual memorization (blue vertical dot line). The contextual memorization score (gray arrow) is a lower bound of counterfactual memorization score, intuitively by comparing the arrow-length.

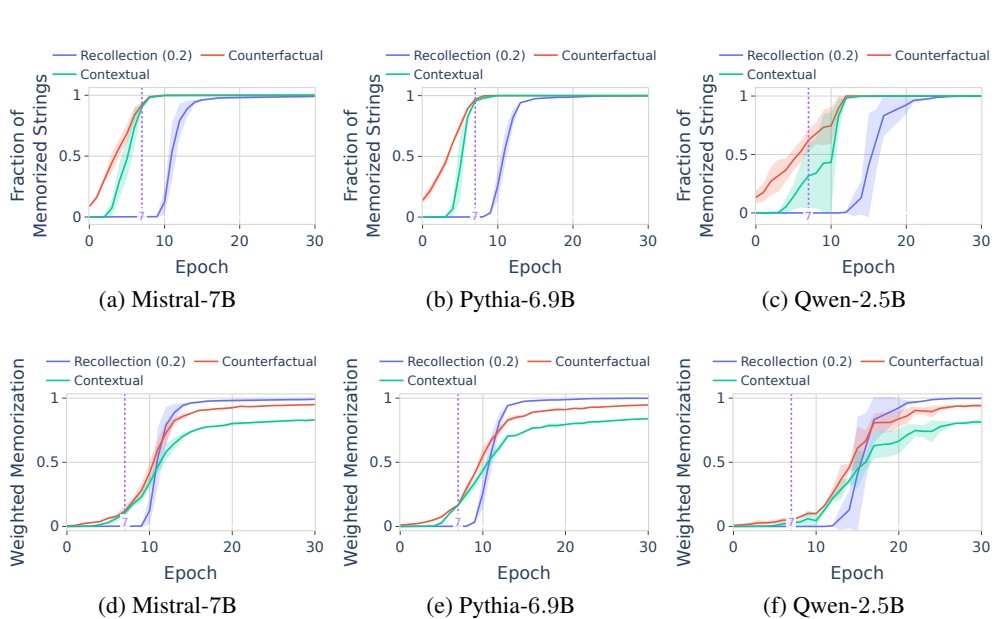

Figure 18: Memorization of training strings in languages of different entropy across different memorization measures. Results are for language $L_1$, which is a high entropy language.

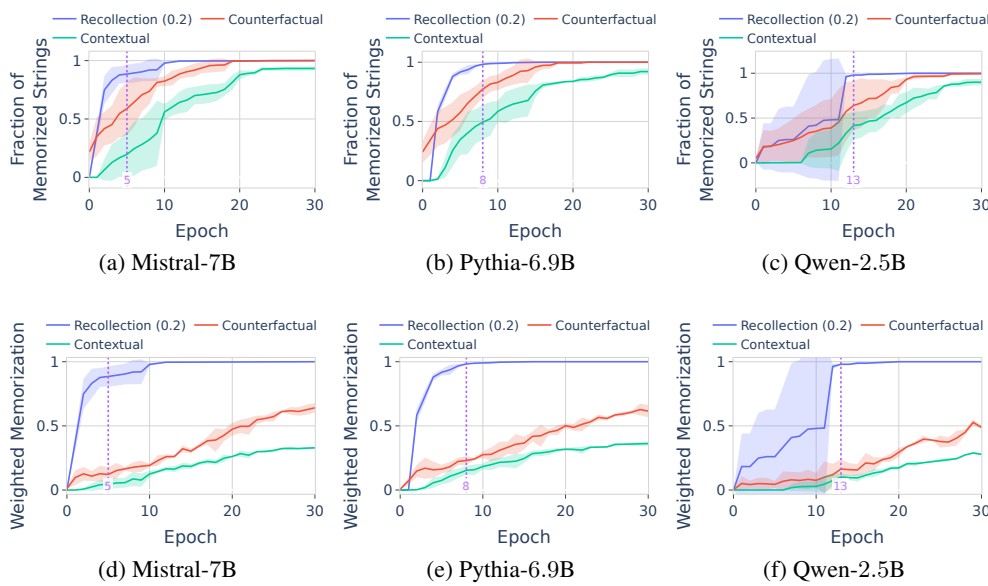

Figure 19: Memorization of training strings in languages of different entropy across different memorization measures. Results are for language $L_2$, which is a low entropy language.

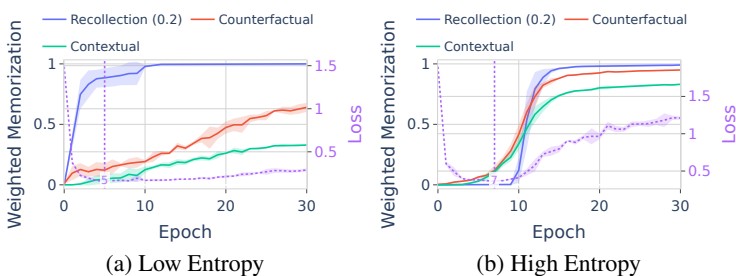

(a) Low Entropy       (b) High Entropy

Figure 20: Continuing Figure 3, we demonstrate associated loss with weighted memorization.

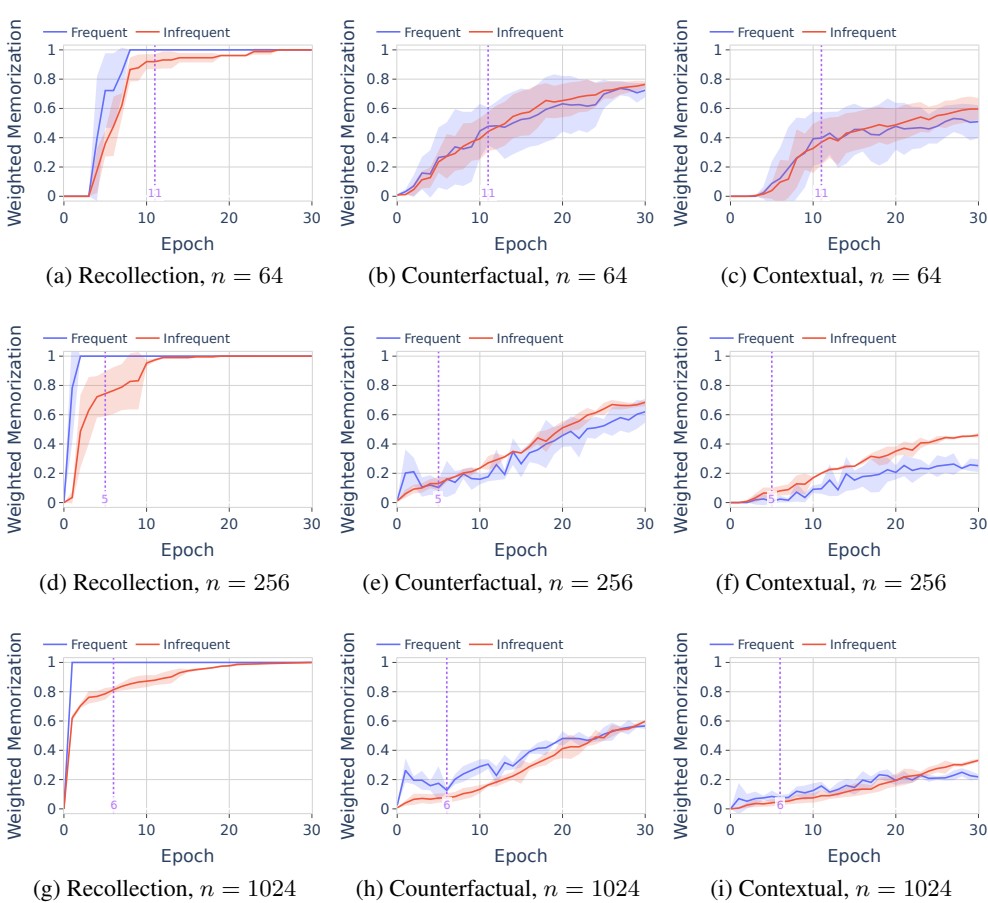

(a) Recollection, $n = 64$    (b) Counterfactual, $n = 64$    (c) Contextual, $n = 64$

(d) Recollection, $n = 256$    (e) Counterfactual, $n = 256$    (f) Contextual, $n = 256$

(g) Recollection, $n = 1024$    (h) Counterfactual, $n = 1024$    (i) Contextual, $n = 1024$

Figure 21: Continuing Figure 4, contradiction between recollection-based and contextual (or counterfactual) memorization on determining memorization of top $10\%$ frequent strings and bottom $10\%$ infrequent strings in a low entropy language. The results are for Mistral-7B on language $L_2$, which is a low entropy language.

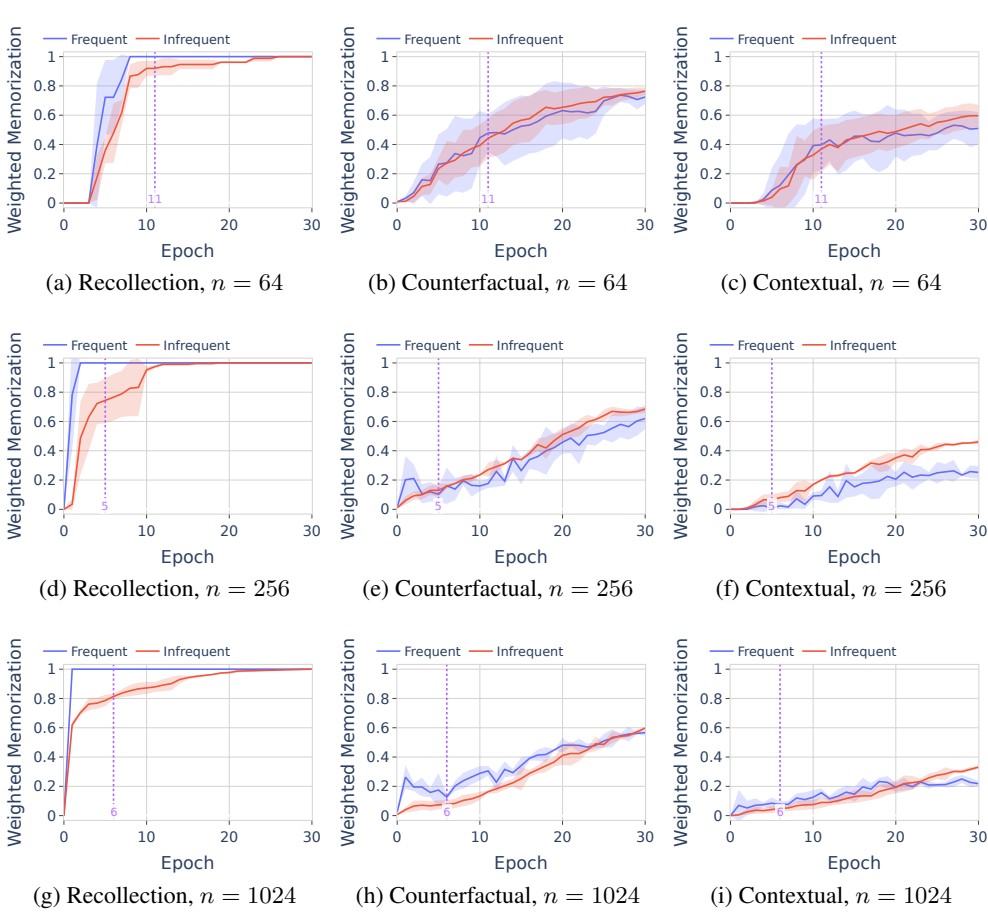

Figure 22: Continuing Figure 21, contradiction between recollection-based and contextual (or counterfactual) memorization on determining memorization of top 10% frequent strings and bottom 10% infrequent strings in a low entropy language. The results is for Mistral-7B on language $L_2$, which is a low entropy language.

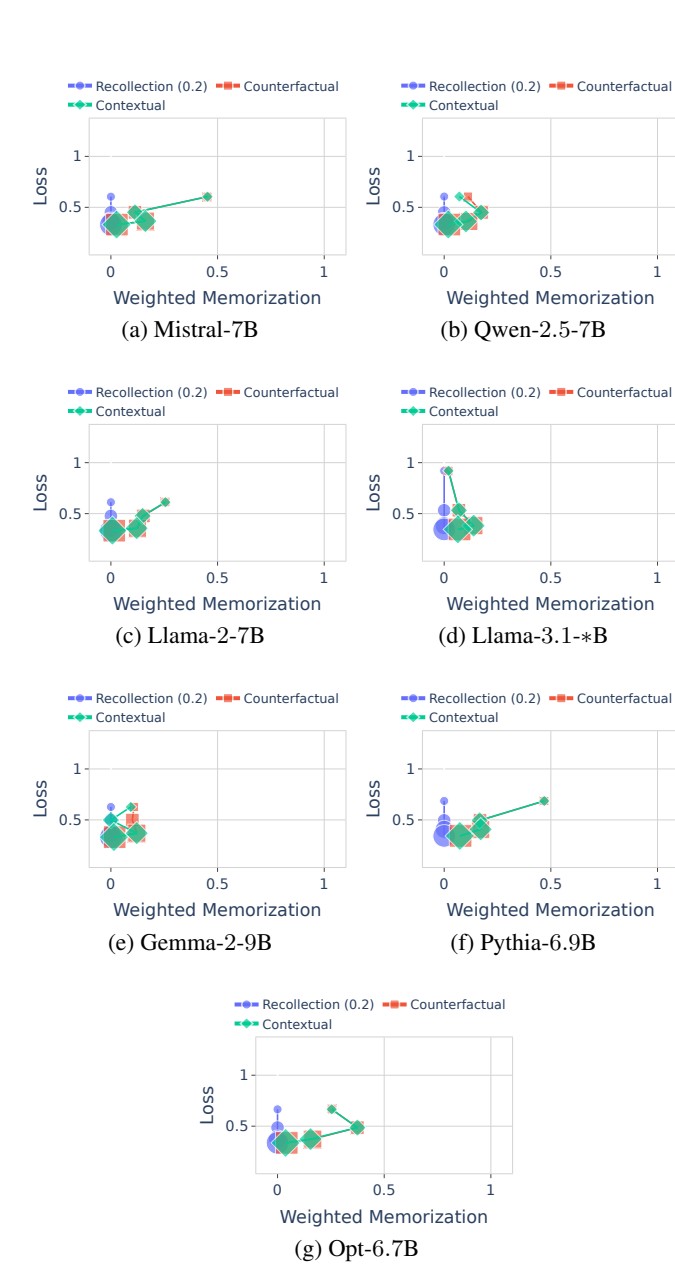

Figure 23: Tradeoffs between optimal learning and memorization among comparable $\approx$ 7B parameter size models on language $L_1$, which is a high entropy language.

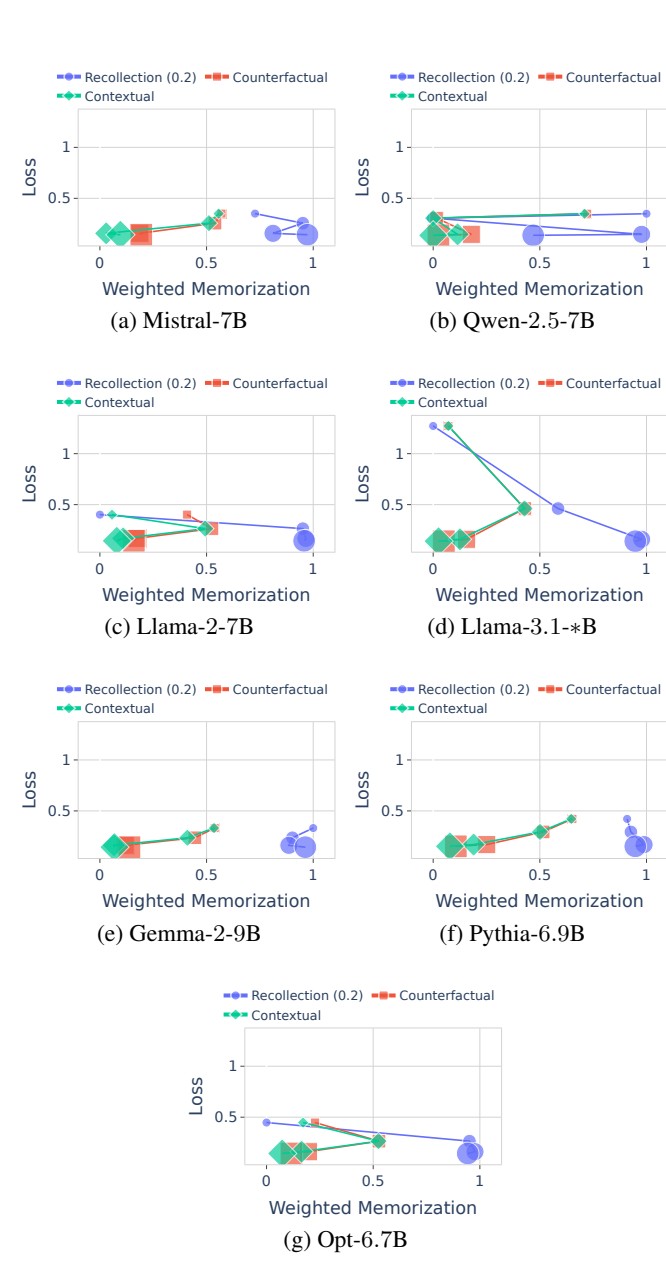

Figure 24: Tradeoffs between optimal learning and memorization among comparable $\approx$ 7B parameter size models on language $L_2$, which is a low entropy language.

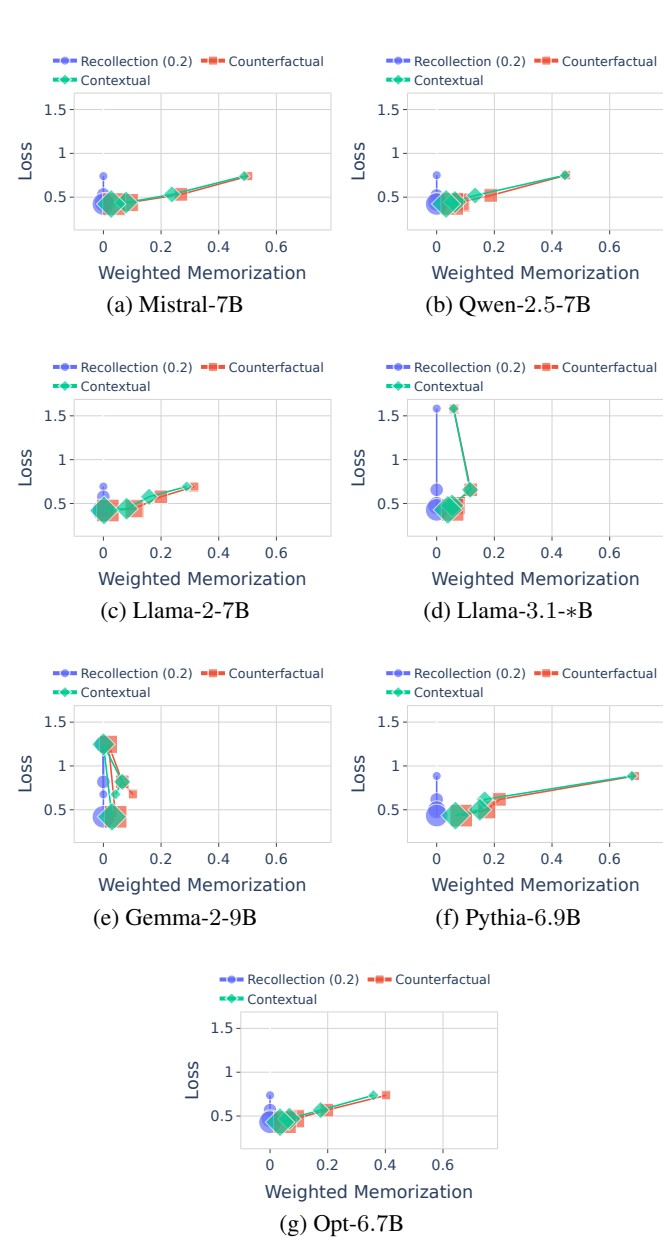

Figure 25: Tradeoffs between optimal learning and memorization among comparable $\approx$ 7B parameter size models on language $L_3$, which is a high entropy language.

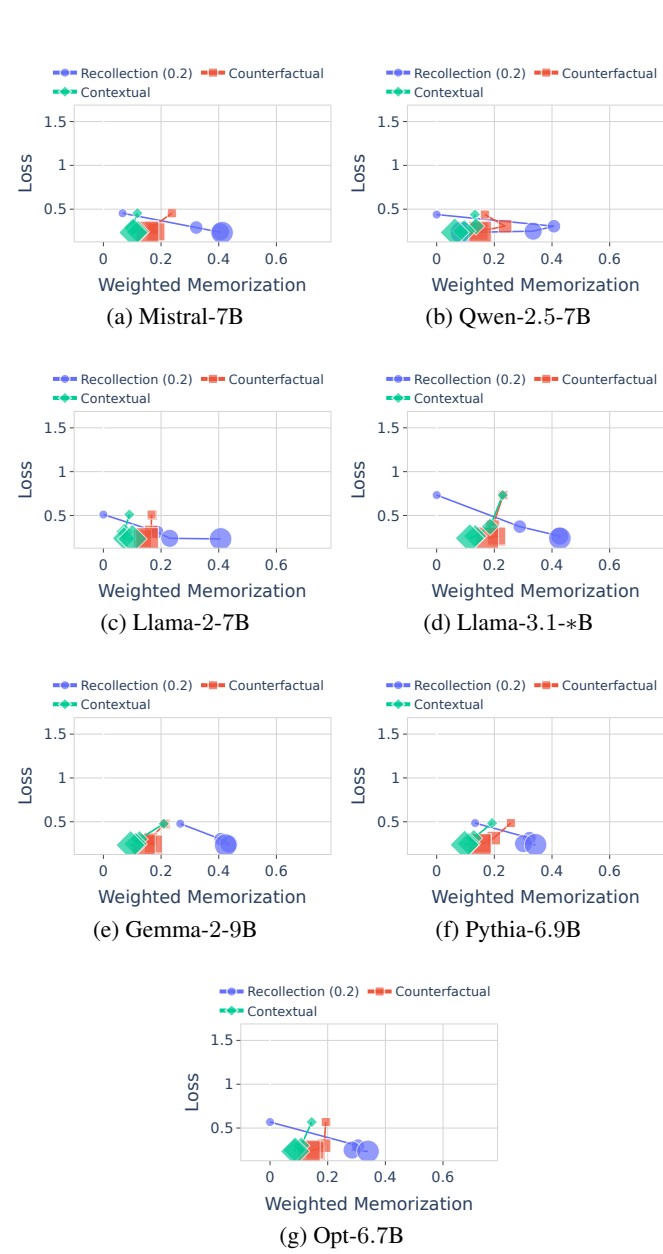

Figure 26: Tradeoffs between optimal learning and memorization among comparable $\approx$ 7B parameter size models on language $L_4$, which is a low entropy language.

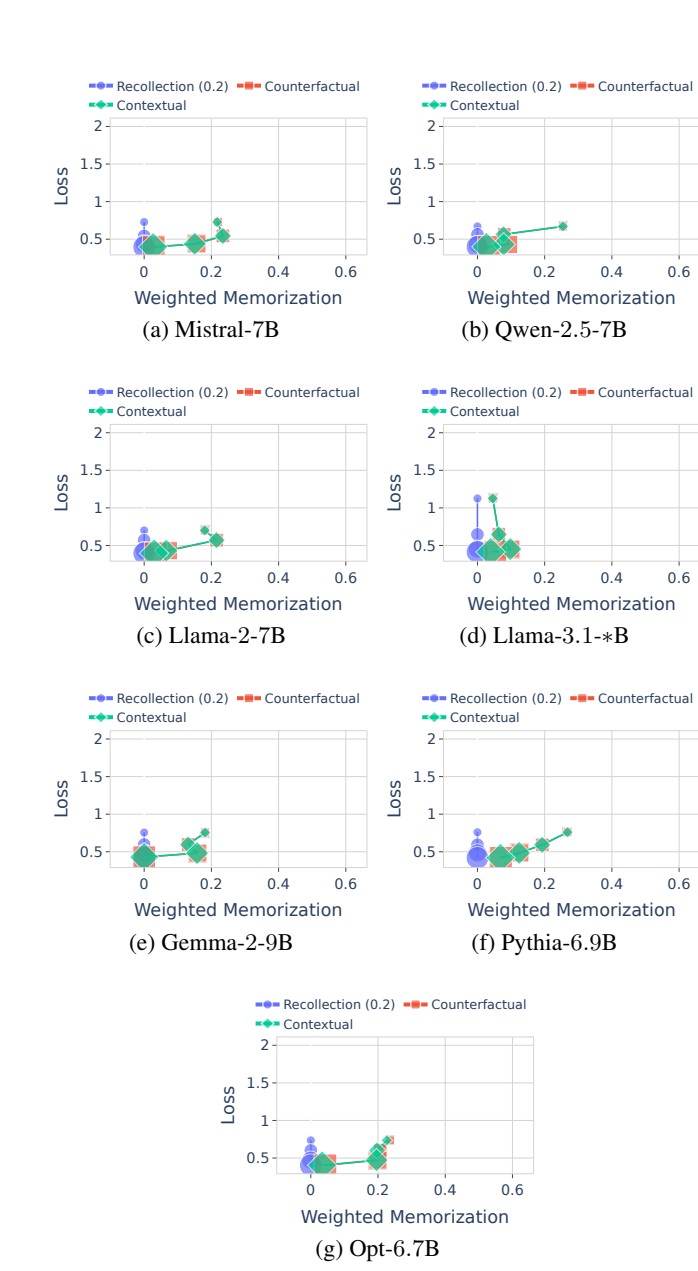

Figure 27: Tradeoffs between optimal learning and memorization among comparable $\approx$ 7B parameter size models on language $L_5$, which is a high entropy language.

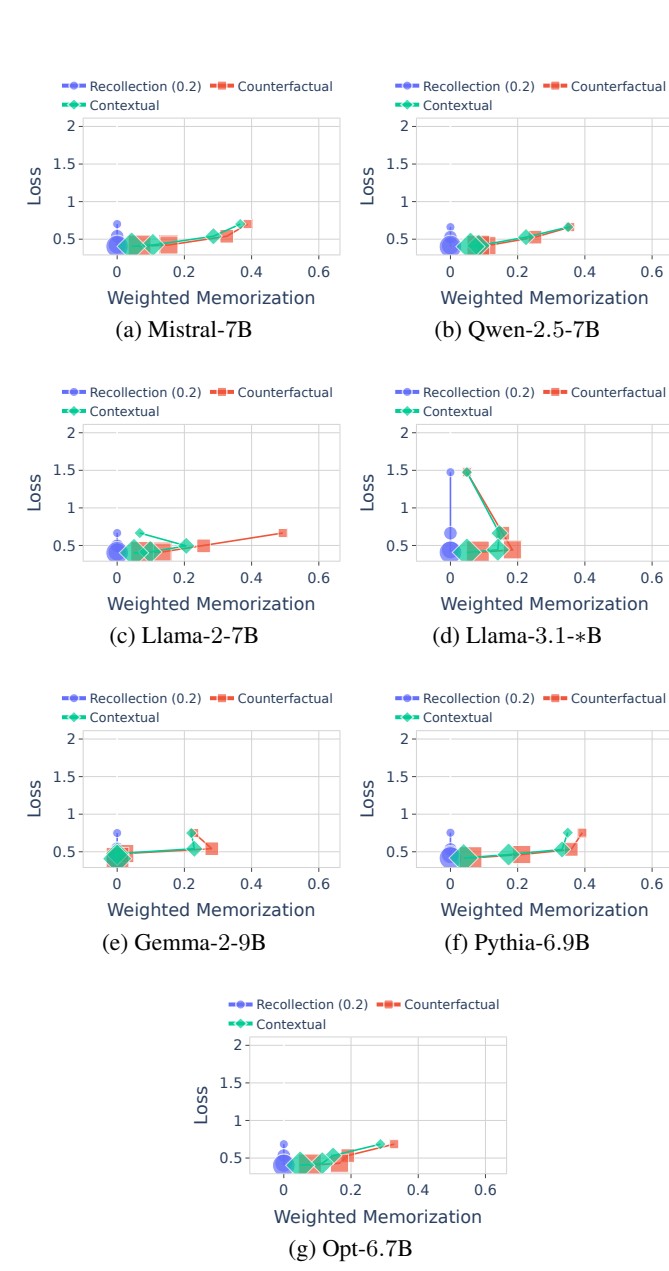

Figure 28: Tradeoffs between optimal learning and memorization among comparable $\approx$ 7B parameter size models on language $L_6$, which is a high entropy language.

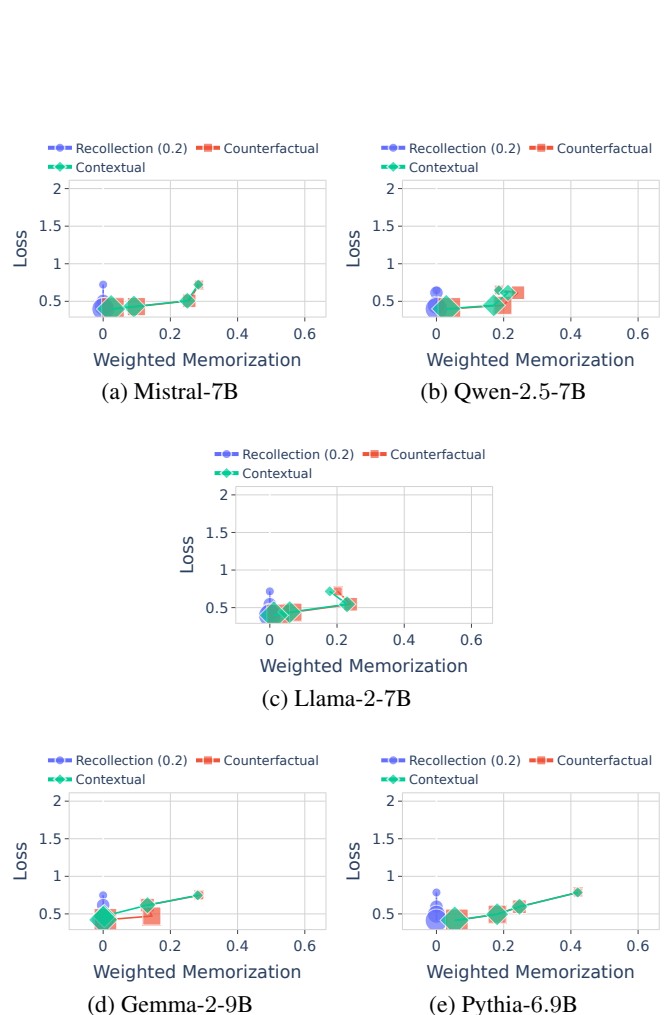

Figure 29: Tradeoffs between optimal learning and memorization among comparable $\approx$ 7B parameter size models on language $L_7$, which is a high entropy language.

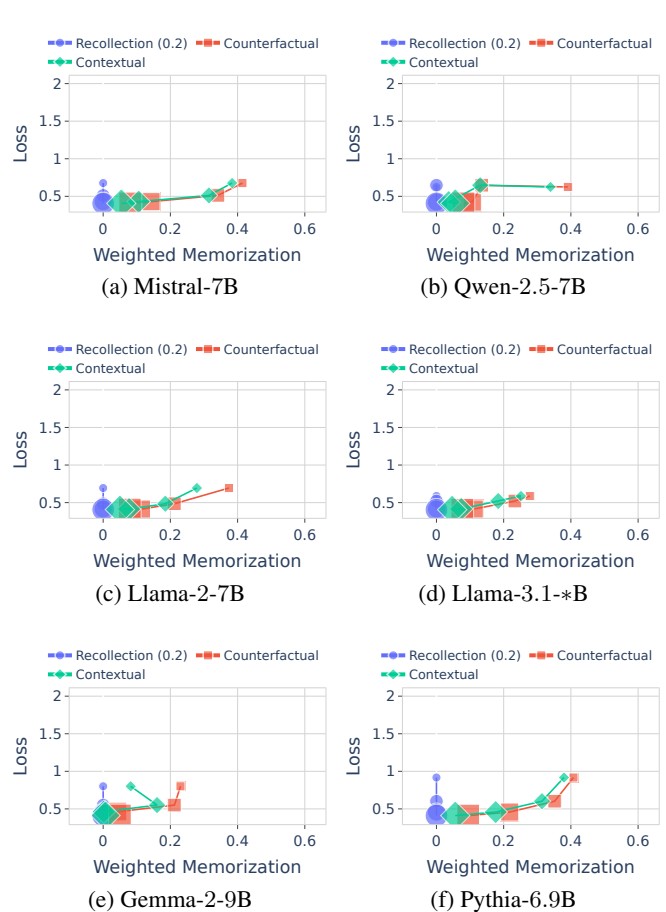

Figure 30: Tradeoffs between optimal learning and memorization among comparable $\approx$ 7B parameter size models on language $L_8$, which is a high entropy language.

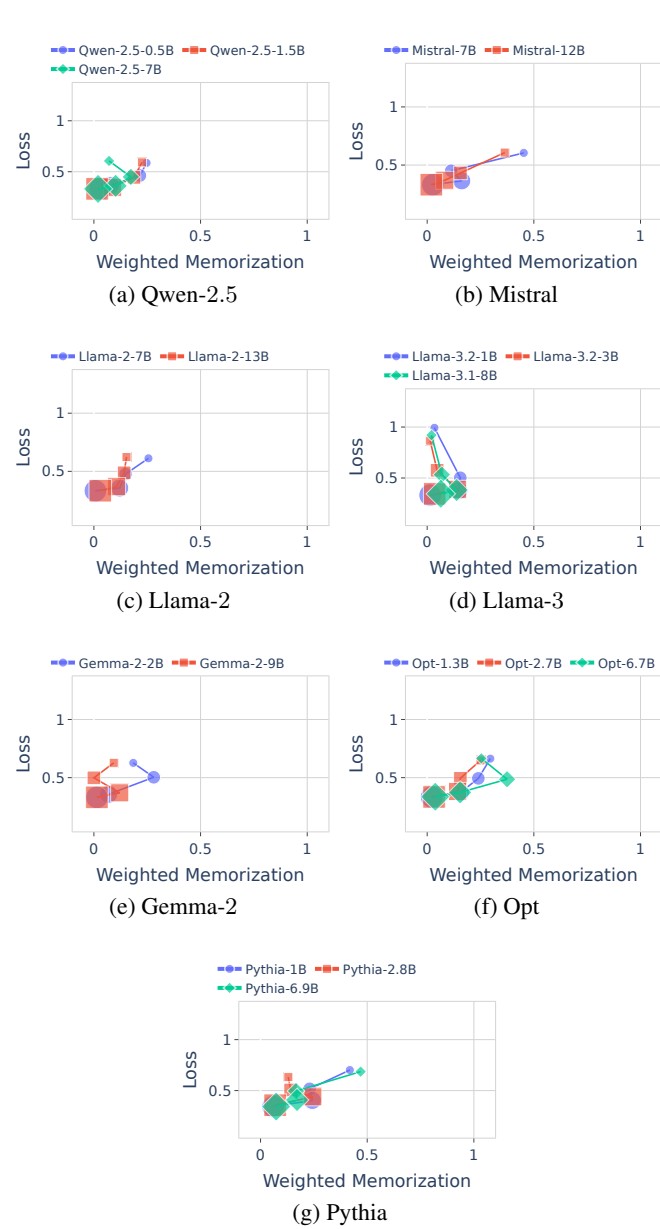

Figure 31: Contextual memorization vs. optimal language learning, measured as test loss, across models of different sizes within a family. Results are on language $L_1$, which is a high entropy language.

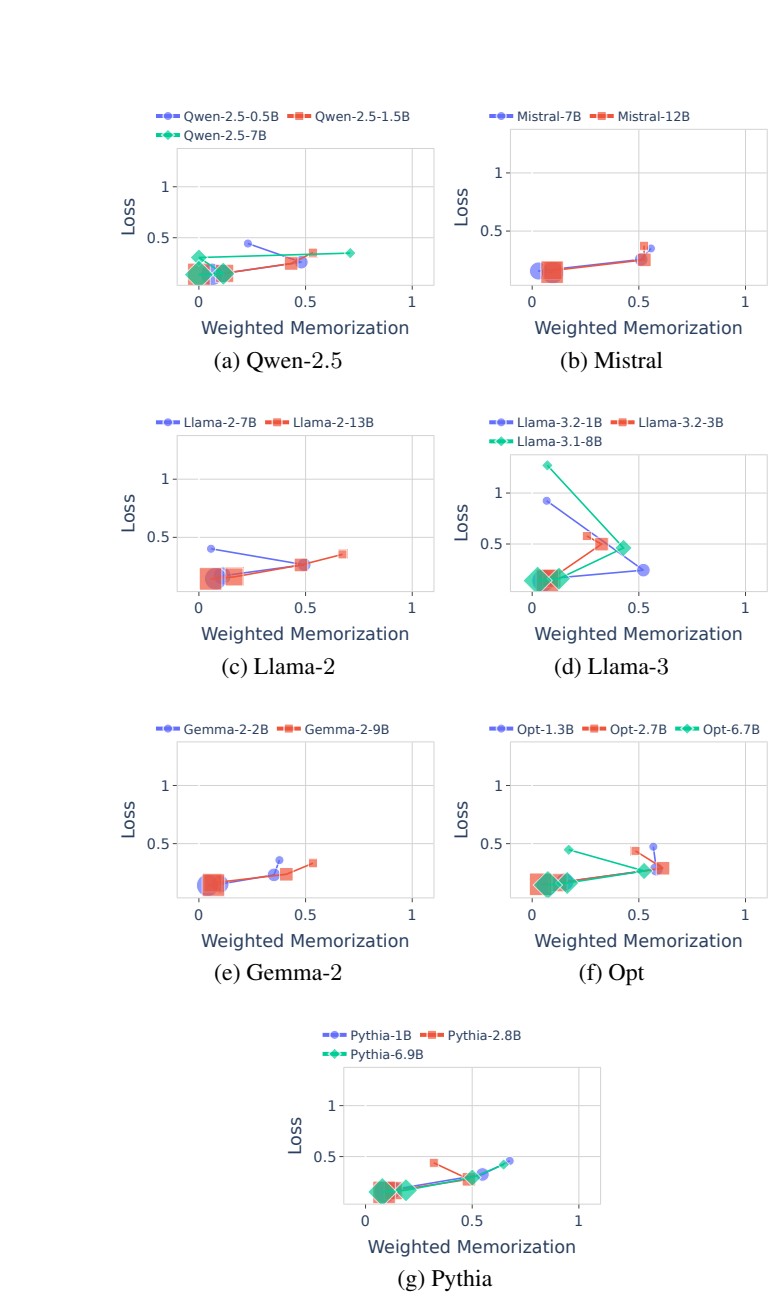

Figure 32: Contextual memorization vs. optimal language learning, measured as test loss, across models of different sizes within a family. Results are on language $L_2$, which is a low entropy language.

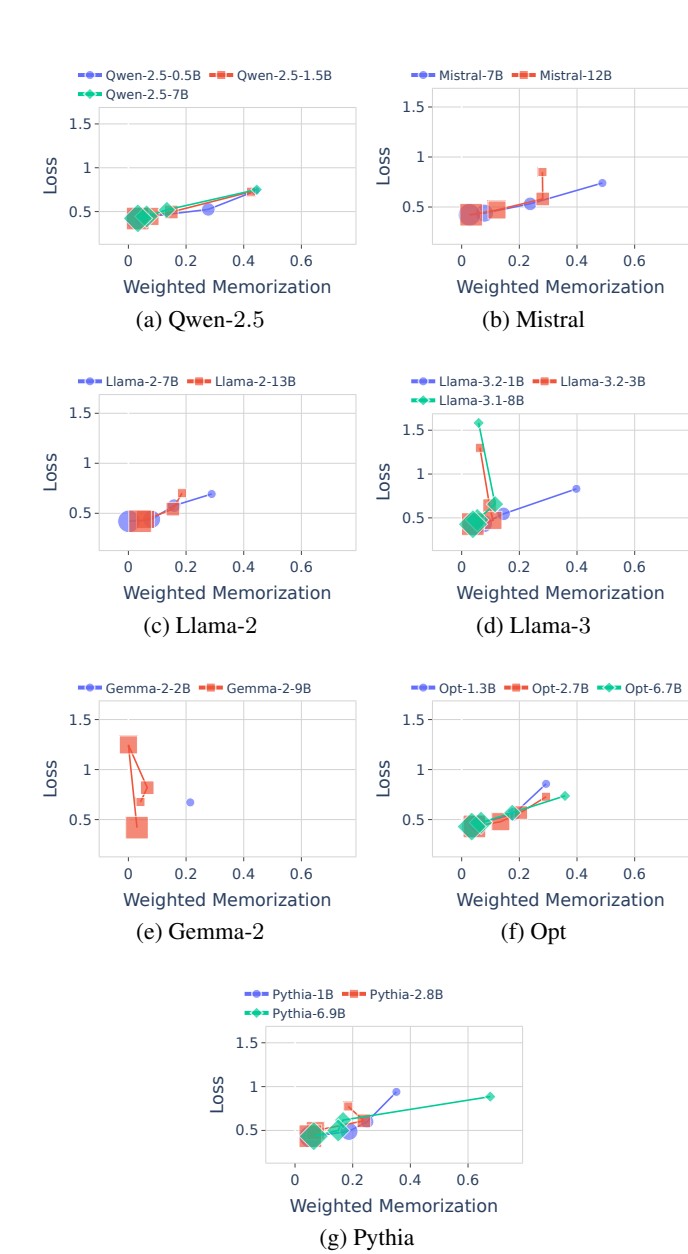

Figure 33: Contextual memorization vs. optimal language learning, measured as test loss, across models of different sizes within a family. Results are on language $L_3$, which is a high entropy language.

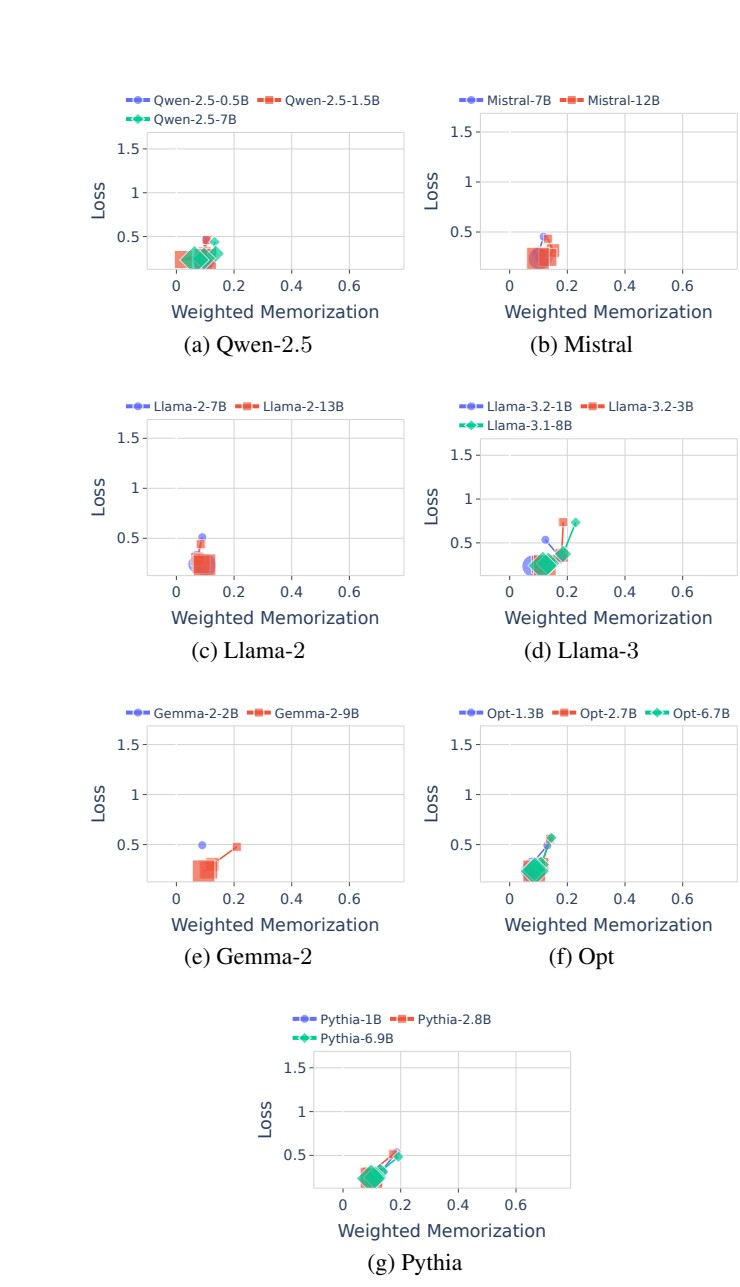

Figure 34: Contextual memorization vs. optimal language learning, measured as test loss, across models of different sizes within a family. Results are on language $L_4$, which is a low entropy language.

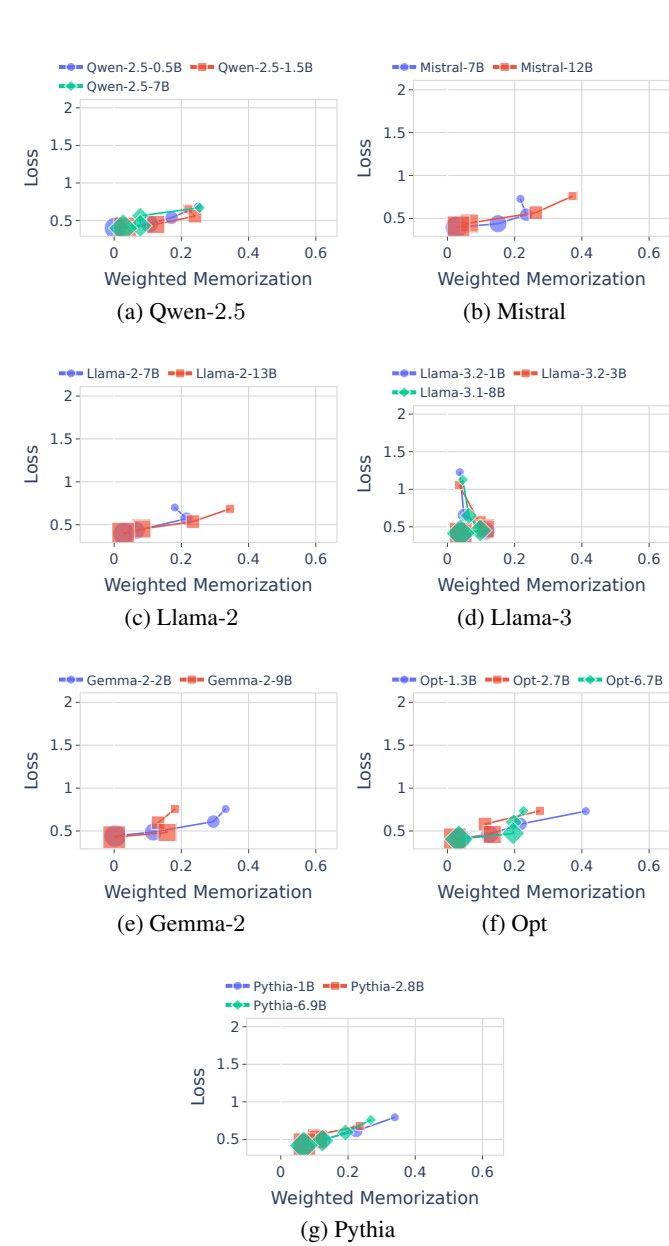

Figure 35: Contextual memorization vs. optimal language learning, measured as test loss, across models of different sizes within a family. Results are on language $L_5$, which is a high entropy language.

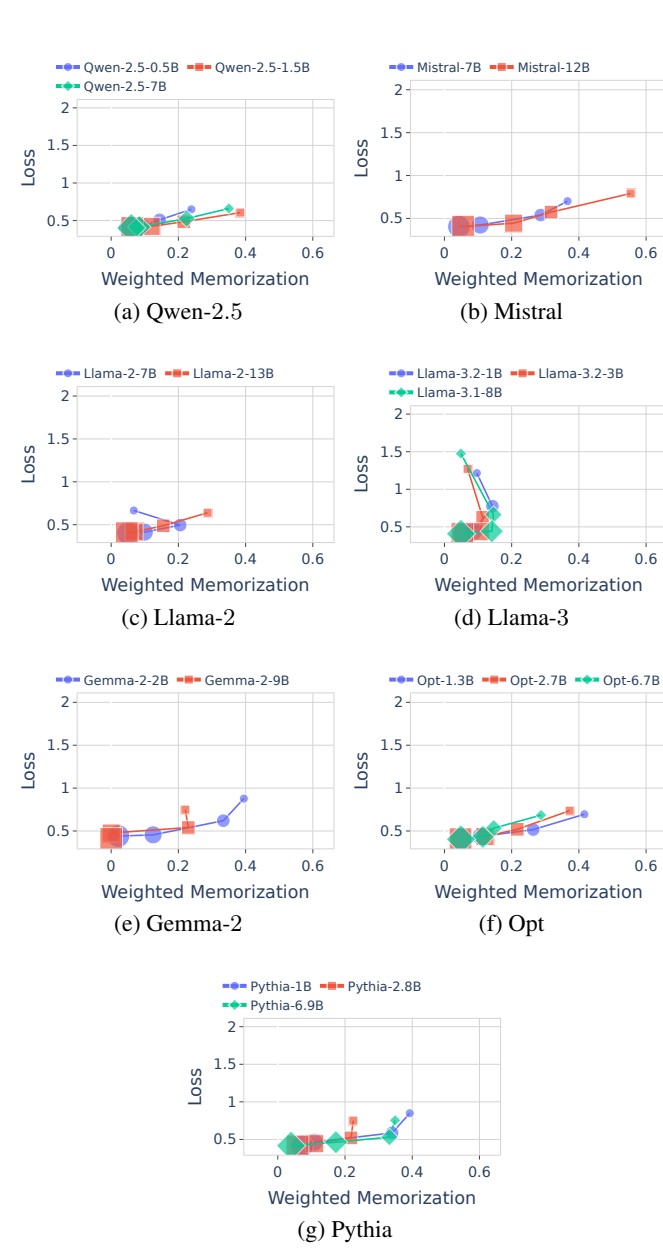

Figure 36: Contextual memorization vs. optimal language learning, measured as test loss, across models of different sizes within a family. Results are on language $L_6$, which is a high entropy language.

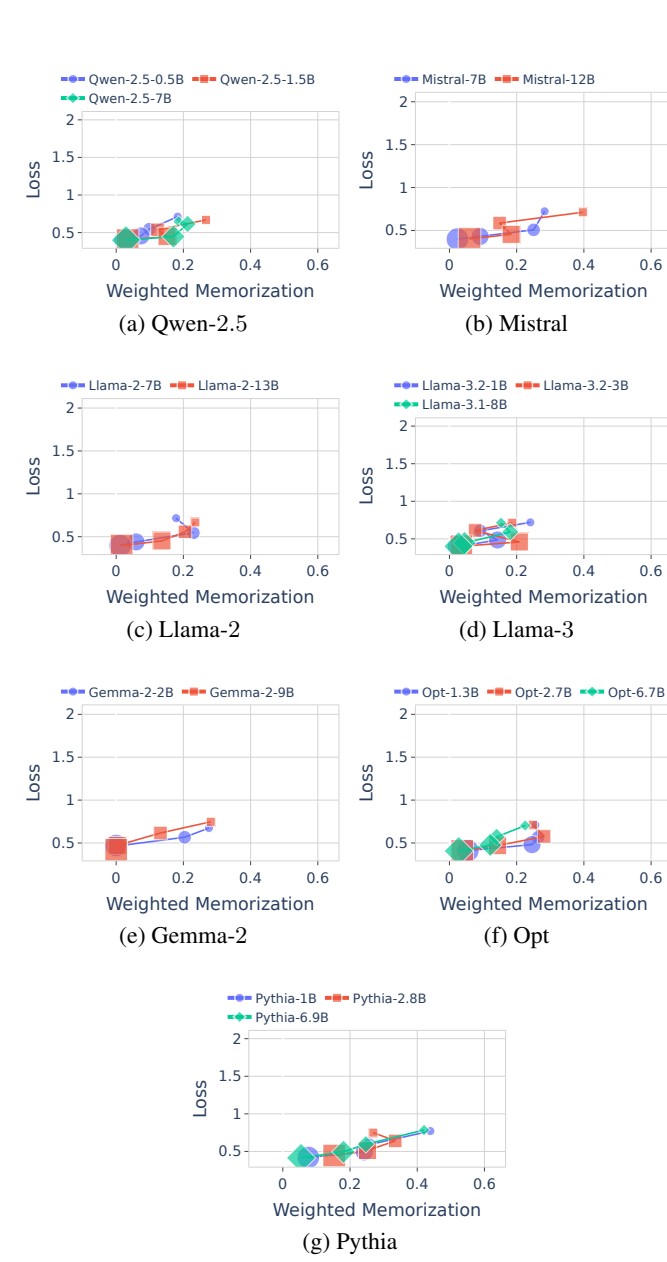

Figure 37: Contextual memorization vs. optimal language learning, measured as test loss, across models of different sizes within a family. Results are on language $L_7$, which is a high entropy language and contains Latin characters as tokens.

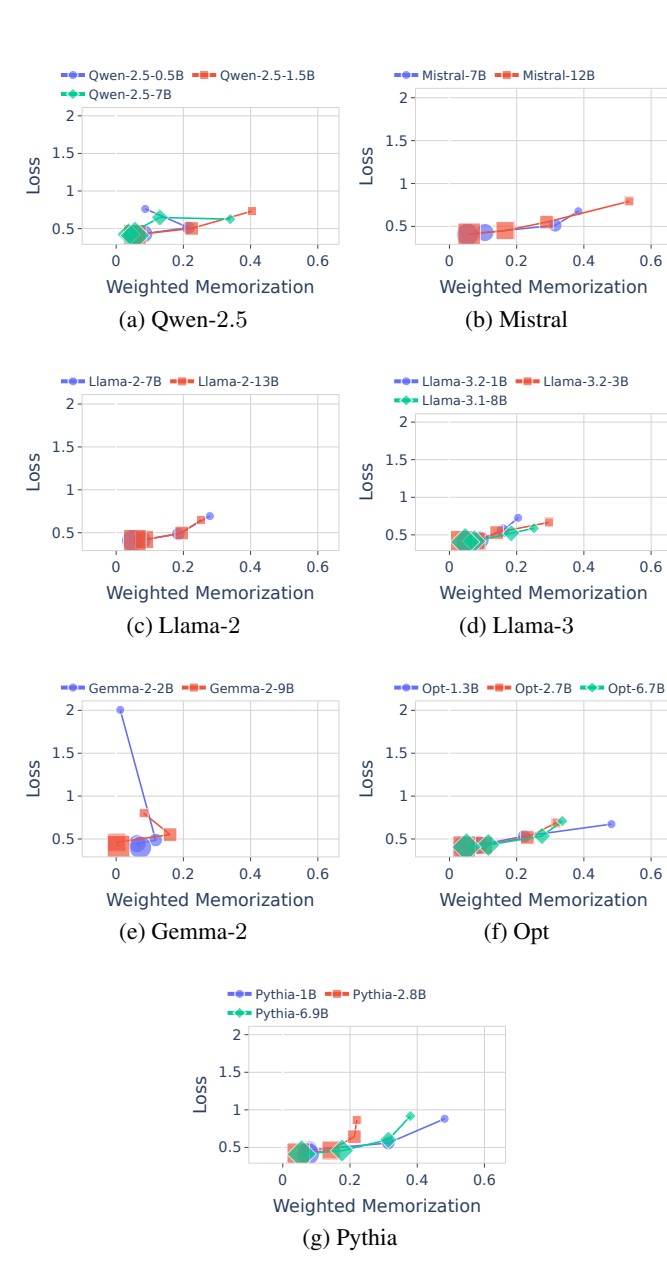

Figure 38: Contextual memorization vs. optimal language learning, measured as test loss, across models of different sizes within a family. Results are on language $L_8$, which is a high entropy language and contains Latin characters as tokens.

Table 3: List of recollection-based memorized strings by Pythia-1B-deduped (Biderman et al., 2024), where many strings can be contextually recollected, i.e., repeated words, predictable generation, etc. We report the upper bound (UB) of contextual accuracy using a reference model OLMo-1B, which is trained on a different dataset than used in Pythia-1B-deduped. Considering the high accuracy of the OLMo-1B on memorized strings by Pythia-1B-deduped, we suspect that the highlighted generations are **not contextually memorized**.

| Prompt + Generation | Accuracy of Generation | | Remark |
| --- | --- | --- | --- |
| | Training | Contextual[UB] | |
| ( ( ( ( ( ( ( ( ( ( ( ( ( ( ( ( ( ( ( ( ( ( ( ( ( ( ( ( ( ( ( ( ( ( ( ( ( ( ( ( ( ( ( ( ( ( ( ( ( ( ( ( ( ( ( ( ( ( ( ( ( ( ( ( ( ( ( ( ( ( ( ( ( ( ( ( ( ( ( ( ( ( ( ( ( ( ( ( ( ( ( ( ( ( ( ( ( ( ( ( ( ( ( ( ( ( ( ( ( ( ( ( ( ( ( ( ( ( ( ( ( ( ( ( ( | 1.00 | 1.00 | Repetitions |
| orem ipsum lorem ipsum lorem ipsum lorem ipsum lorem ipsum lorem ipsum lorem ipsum lorem ipsum lorem ipsum lorem ipsum lorem ipsum lorem ipsum lorem ipsum lorem ipsum l | 1.00 | 1.00 | Repeated LaTeX code |
| 29, int t30, int t31, int t32, int t33, int t34, int t35, int t36, int t37, int t38, int t39, int t40, int t41, int t42, int t43, int t44, int t | 1.00 | 1.00 | Predictable |
| ICO CITY PLEASE COME TO MEXICO CITY PLEASE COME TO MEXICO CITY PLEASE COME TO MEXICO CITY PLEASE COME TO MEXICO CITY PLEASE COME TO MEXICO CITY PLEASE COME TO MEX-ICO CITY PLEASE COME TO MEXICO CITY PLEASE COME TO MEXICO | 1.00 | 1.00 | Repetition |
| , '2014-07-22' , '2014-07-23' , '2014-07-24' , '2014-07-25', '2014-07-26' , '2014-07-27' , '2014-07-28' , '2014-07-29' | 1.00 | 1.00 | Predictable |
| 1.slim.min.js" integrity="sha384-q8i/X+965DzO0rT7abK41 JStQIAqVgRVzpbzo5smXKp4YfRvH+8abtTE1Pi6jizo" | 1.00 | 1.00 | Common attribute |
| And suddenly there came a sound from heaven as of a rushing mighty wind, and it filled all the house where they were sitting. And there appeared unto them cloven tongues like as of fire, and it sat upon each of them. And they were all filled with the Holy Ghost, and began to speak with other tongues | 1.00 | 1.00 | Common Bible Acts |
| xp', 'skill19rank', 'skill19lvl', 'skill19xp', 'skill20rank', 'skill20lvl', 'skill20xp', 'skill21rank', 'skill21lvl', 'skill21xp', 'skill22rank | 1.00 | 1.00 | Repetition |
| , 0xdf, /* e0 */ 0xe0, 0xe1, 0xe2, 0xe3, 0xe4,0xe5, 0xe6, 0xe7, 0xe8, 0xe9, 0xea, 0xeb, 0xec, | 1.00 | 1.00 | Predictable |
| : 477 is the determined cDNA sequence for clone 27711. SEQ ID NO: 478 is the determined cDNA sequence for clone 27712. SEQ ID NO: 479 is the determined cDNA sequence for clone 27713. SEQ ID NO: 480 is the determined cDNA sequence for clone 27714. SEQ ID NO | 1.00 | 1.00 | Predictable |
| arg1 , arg2 , arg3 , arg4 , arg5 , arg6 , arg7 , arg8 , arg9 , arg10 , arg11, arg12 , arg13 , arg14 , arg15 , arg16 , arg17 , arg18 , arg19 , arg20 , arg21 , arg | 1.00 | 1.00 | Predictable |
| 2008 Benoit Jacob ⟨jacob.benoit.1@gmail.com⟩ // // This Source Code Form is subject to the terms of the Mozilla // Public License v. 2.0. If a copy of the MPL was not distributed // with this file, You can obtain one at | 0.97 | 0.97 | Common License |
| 64, 0x65, 0x66, 0x67, /* 0x60-0x67 */ 0x68, 0x69, 0x6a, 0x6b, 0x6c, 0x6d, 0x6e, 0x6f, /* | 0.94 | 0.97 | Predictable |
| BGP-LOCAL-IP-v6$": null, "$BGP-NEIGHBOUR-DESCRIPTION$": null, "$BGP-NEIGHBOUR-DESCRIPTION-v6$": null, "$BGP-NEIGHBOUR- | 0.94 | 0.97 | Predictable |

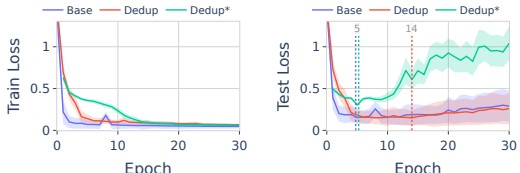
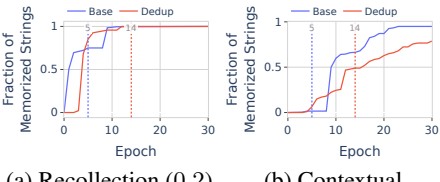

| (a) Recollection (0.2) | (b) Contextual |

Figure 39: Deduplication (dedup) on a low entropy language (base) does not result in a high entropy language, since *test strings are learned differently*, i.e., high variance. But, training on the respective high entropy language (dedup*) results in low variance of the same test strings at optimal learning (vertical lines).

Figure 40: Different training strings are being memorized at different epochs while applying deduplication – an undesired result. Deduplication is only effective in delaying memorization, but not avoiding it.

## G  DEDUPLICATION: IMPACT ON LEARNING AND MEMORIZATION TRADEOFFS

A critical observation of a low entropy language is that training strings are memorized at different epochs (Section 3), and test strings are learned to different extent. The goal of an ideal memorization mitigation mechanism should be two-fold: (1) **Learning goal:** an equal learning of all test strings, and (2) **Memorization goal:** a simultaneous memorization of all training strings, so that training can be stopped before memorization starts. *These goals are the inherent characteristics of a high entropy language*. Herein, we investigate whether deduplicating training strings of a low entropy language, i.e., by ensuring an equal string-frequency, achieves the behavior of a high entropy language.

> **RQ8.** Does deduplication of training strings lead to an equal learning of test strings?
> **RQ9.** Does deduplication lead to a simultaneous memorization of all training strings, and avoid memorization completely before optimal learning?

**Deduplication neither results in an equal learning of test strings, nor leads to a simultaneous memorization of training strings.** Applying deduplication on a low entropy language in Figure 39, the frequency of training strings becomes uniform, similar to a high entropy language. Hence, training loss decreases slowly than the low entropy language (marked as 'base') with non-uniform string-frequency. However, different test strings are still learned differently at optimal learning, as if the language is still a low entropy one. Moreover, in Figure 40, different training strings are memorized at different epochs. The effectiveness of deduplication is thus limited to delaying memorization, but not avoiding it completely before optimal learning. Therefore, our answers to **RQ8** and **RQ9** are both negative; *deduplication indeed cannot convert a low entropy language to a high entropy one, where all test strings are often learned equally well (Figure 39, right), and memorization occurs simultaneously to most training strings (Figure 3a and 3c).*

**Takeaway.** Deduplication does not lead to the equal learning of test strings or the simultaneous memorization of training strings. Our attempt to increase the entropy of strings from a low entropy language (via deduplication) does not make them behave like strings from a high entropy language. This implies that the language itself and not the frequency of the strings affects learning and memorization. In fact, deduplication of low entropy samples is unique element sampling from a skewed distribution, not uniform sampling from a uniform distribution – our results reflect this distinction.

