# OpenReview forum: "Learning without Memorizing Considered Infeasible: Rethinking Memorization in LLMs"
_ICLR.cc/2026/Conference — Submitted to ICLR 2026_

### Official Review · Reviewer_Qw3D · 2025-10-16

**Soundness:** 2
**Presentation:** 1
**Contribution:** 2
**Rating:** 2
**Confidence:** 3

**Summary:**

The paper is concerned about memorization in large language models (LLMs) and its implications for privacy and learning. It critiques existing measures of memorization, which are primarily recollection-based and counterfactual, arguing that they often conflate memorization with contextual learning. The authors propose a new measure called "contextual memorization" that better distinguishes between these two phenomena. They conduct extensive experiments with multiple LLMs across various formal languages. From these experiments, they claim two main findings: (1) learning a language without any memorization is infeasible, and (2) current assessments of LLM memorization are likely exaggerated. Particularly, they find that many instances of privacy risk study where LLMs memorize training under recollection-based measures can be explained by contextual learning instead.

**Strengths:**

- The paper considers an important and timely topic, given the increasing use of LLMs and concerns about privacy and memorization.
- The paper presents many results, including a few quite interesting ones. For instance, Figure 2 compares different memorization measures and shows that they can lead to different conclusions about when a string starts to be memorized; Figure 5 shows that training data deduplication does not reduce the band of epochs when strings in a low entropy language are memorized.
- Section 3.1 is helpful in understanding the differences between various memorization measures considered in this paper.
- The empirical evaluation is extensive, covering 18 LLMs across 6 model families and multiple formal languages.

**Weaknesses:**

- It is difficult for me to fully follow the paper. This is partly because the paper presents many results, but connects them only loosely. The technical sections (Sections 3, 4, and 5) are three separate works, among which, seven disjoint RQs (I assume this means research questions) are asked and answered. These research questions, as well as their respective findings, although valuable on their own, do not seem to all serve the same purpose of validating the main claims of the paper. For example, RQ5, "Increasing training dataset size improves optimal learning, but do we risk memorizing more training strings (due to more repeated strings)?" is interesting, but it is not clear how it relates to the main claims of the paper. The overall structure and organization of the paper could be improved a lot.
- Some of the claims are not substantiated well. Since the paper proposes the new measure of contextual memorization, it is important to establish cases where this measure is more accurate than existing measures, especially the very similar counterfactual memorization measure. I assume Lines 273 to 288 are trying to do this, but I don't find the arguments very convincing. It is true that contextual memorization imposes a stricter threshold than counterfactual memorization, and it is also true that the motivations and the empirical behaviors for the two measures are different. But it is not clear why these differences make contextual memorization a better measure at least in some cases.
- A non-negligible portion of the results are presented in unclear ways. I'll give two examples *among many*.
    - Figure 1 looks like a real run of training instead of an illustration, but its related settings are never mentioned throughout the paper. Should I consider this figure a general trend that one should always expect to see? Otherwise, why is it chosen to be here?
    - In Lines 445 to 458, what exacly was the experiment setting? What does "accuracy" mean in this experiment? How do you generate the reference model $M_{ref}$? "If a string memorized by $M$ is generated by $M_{ref}$ with equal or higher recollection, it is unlikely to be contextually memorized." - Is that validated somewhere? How is "accuracy" of $M_{ref}$ an upper bound of the optimal contextual accuracy?

Overall, although the paper presents a lot of results, with some interesting ones, it is difficult for me to follow the paper and be convinced by its main claims. I think the paper needs a major revision to improve its structure and organization, as well as to better substantiate its claims.

**Questions:**

- Most of the figures contain error bars, how are those computed? Does that come from multiple runs of training? If so, how many runs?
- What is the amount of tokens used for training a model for the formal languages you use?

### Minor comments

- Line 166: "Gemma (Team et al., 2024)" seems problematic with the author name.
- Line 207: I recommend writing $D \ni s$ instead of $s \in D$, since $s$ is fixed and $D$ is a set that varies.
- Line 329: I believe the correct notation is $\mathbb{E}_{s \sim D} [\cdot]$ instead of $\mathbb{E}_{s \in D} [\cdot]$.

---

> ### Author Response · Authors · 2025-11-21
>
> We thank the reviewer for appreciating the paper. We address the questions below.
>
> **Structure and organization of the paper.** The primary motivation of the paper is to study memorization in LLMs from two perspectives: learning risks and privacy risks. To this end, we make a deeper contrast between different measures of memorization, and propose contextual memorization, which is the only measure focusing on learning risks. The three main sections have the following objectives: Section 3 studies the disagreement of different memorization measures, Section 4 studies the relation between memorization and learning, and Section 5 studies the relation between memorization and privacy risks. In the final version of the paper, we make the connection more coherent.
>
> **Difference between contextual and counterfactual memorization.** We do not make a subjective opinion of which measure between contextual and counterfactual memorization is better. In fact, we make a note that both are better than recollection-based memorization, which fails to differentiate between recollection due to memorization or due to learning.
>
> Contextual memorization and counterfactual memorization are proposed to detect learning and privacy risks, respectively. The privacy aspect may tolerate a *false positive* of classifying a string as memorized, but the learning aspect demands the necessary condition for memorization in order to avoid any false positive. Contextual memorization being stricter than counterfactual memorization results in lower false positive, which may be appreciated by the learning perspective but not from the privacy perspective, making it subjective.
>
>
> **Figure 1.** The related setting of Figure 1 is discussed in Section 3.2. The figure is meant to depict a *general* trend, where with training epochs, the training loss of a string decreases. The contextual or counterfactual loss is shown using the dash-dot line, which denotes the loss of the same string when it is excluded from training.
>
>
> Since the LLM acquires contextual understanding of the language, contextual loss decreases initially for a few epochs. But the loss increases when the LLM begins memorizing the string by locally over-fitting on it. This observation motivates us to propose contextual memorization, where the memorization threshold is the lowest contextual loss.
>
>
> **Lines 445 to 458**. In Lines 445 to 458, our aim to find out whether a string $s$ claimed to be memorized by the recollection-based measure by model $M$ can be explained away (or filtered out) by the contextual memorization. To this end, we must know the optimal contextual threshold of $s$ to determine contextual memorization, which requires retraining $M$ without $s$. Since retraining is infeasible in practice, we resort to a proxy of contextual threshold using a reference model $M_{\text{ref}}$.
>
> First, we realize that since $s$ is claimed to be memorized by $M$, $s$ should be within the training data of $M$. Next, we make a *hypothesis* that if $M$ and $M_{\text{ref}}$ do *not have any common training data*, and $M_{\text{ref}}$ generates $s$ with equal or higher recollection than $M$, then the only explanation of $M_{\text{ref}}$ generating $s$ is based on contextual learning, i.e., $s$ contains predictable tokens that do not require memorization. Therefore, $s$ is *unlikely* contextually memorized by both $M$ and $M_{\text{ref}}$, where the recollection of $s$ by $M_{\text{ref}}$ is the contextual threshold.
>
>
> In implementation, we use *accuracy* as the measure of recollection, following prior works. Furthermore,  we cannot guarantee disjointness of training data of $M$ and $M_{\text{ref}}$, and hence, the accuracy of $s$ reported by $M_{\text{ref}}$ cannot be the exact contextual accuracy of $s$, but an overestimation (or upper bound) of it.
>
> **Error bars.** To compute the error bars, we consider three random sampling of training strings from the formal language (Line 168).
>
> **Tokens statistics.** For language $L_1$, the average string length is 72 tokens (string length distribution of other languages are in Figure 11). When the training size is $ n = 1024 $, a single epoch (one pass) of training means training on $72 * 1024$ tokens. We repeat training for $ 30 $ epochs, resulting in $ 72 * 1024 * 30 = 2211840 $ tokens per training.
>
> We will address minor comments in the final version of the paper.

---

> > ### Comment · Reviewer_Qw3D · 2025-11-25
> >
> > I would like to thank the authors for their rebuttal.
> >
> > **For structure.** The authors' determination of making the paper more coherent in the next version is appreciated.
> >
> > **For unsubstantiated claims.** I still think that the authors need to identify concrete cases where the proposed measure is more accurate (or more desirable) than existing measures in a convincing way.
> >
> > **For presentation.** I hope the authors understand that for the presentation concerns, it is important for readers reading the paper itself to be able to understand relevant parts.
> >
> > **Error bars.** Line 168 in the current pdf does NOT contain relevant descriptions.
> >
> > For the paper in its current form, I maintain my original rating.

---

### Official Review · Reviewer_w7jk · 2025-10-26

**Soundness:** 1
**Presentation:** 1
**Contribution:** 2
**Rating:** 2
**Confidence:** 3

**Summary:**

The authors
- propose a minor variant of counterfactual memorization they term "contextual memorization"
- compare recollection-based memorization and counterfactual memorization to their new contextual memorization
- use these different notions of memorization to understand questions like how quickly strings are memorized during learning and whether reported memorized strings contain privacy-sensitive PII

**Strengths:**

It's clear that the authors have put in good effort towards this paper.

**Weaknesses:**

## Title:

- I’m not keen on the phrasing of “considered infeasible”. Considered by who? Based on reasonable evidence and meaningful evidence? There is ambiguity concerning whether you (the authors) are advocating this or attacking this?

## Abstract

- “Measures of memorization ignore optimal learning concerns” -> Some measures do, sure, but others don't. I don’t see papers like “Scaling Laws and Interpretability of Learning from Repeated Data” (https://arxiv.org/abs/2205.10487), which explicitly shows in the context of language modeling that memorizing a subset of data can be harmful. I’m not affiliated, but it seems directly relevant to showing that memorization is harmful.
- nit: What does “before” mean in the sentence “... that captures LLMs tendency to locally overfit some strings in the data before during multiple epochs of training.”? Before what?
- nit: Stylistically, I don’t like the newline character in the middle of the abstract. The ICLR instructions are clear: “The abstract must be limited to one paragraph.”
- “Second, we find that when LLMs learn a language optimally, they inevitably end up memorizing some portions of the training data” -> I think to myself, surely these authors will cite the famous Feldman paper (https://arxiv.org/abs/1906.05271) about how memorization is necessary for achieving close-to-optimal generalization error? But I couldn’t find a reference for this paper

## Introduction

- I’m a fan of conceptual schematics. It might be helpful to the reader to add a conceptual subfigure to communicate what contextual memorization is.
- Figure 1 is too tiny.
- Figure 1 doesn’t label in the legend what the dashed and dotted blue and red lines are.
- Figure 1’s caption doesn’t provide adequate information. The reader does not know which models these are, what the data is, what the loss is measured on, etc. If this is meant to be a schematic, it should be labeled as such.
- The thought experiment doesn’t seem to crisply communicate the essence of contextual memorization. I actually think a simpler example from privacy makes more sense. For example, a model may produce PII (e.g., credit card info) but that same PII appears earlier in the sequence, so has the model “memorized” the PII or is it just referring back a page?
- The paragraph “Comparing with Counterfactual Measures” and Figure 1 seems focused on separating contextual from counterfactual memorization. I’m not quite sure that’s a strategic move? Your focus should be on making sure the reader understands your contextual memorization, not a comparison against counterfactual memorization.
- After rereading the paper, I feel like Counterfactual Memorization distracts from (what I understand is) your main point that in-context learning can appear as “memorization” under current metrics.
- I’m unclear about what the difference is between in-context learning and contextual memorization.
- I’m also amazed that (as best as I can tell) “in-context learning” (ICL) does not appear in your manuscript and no ICL papers have been cited. Am I misunderstanding contextual memorization - does it have no connection to ICL?
- The definition of Contextual Memorization doesn’t appear till Page 5. Since this notion is so central to your work, I think it should be stated much earlier.

## Section 2

- Lines 133-134 “prior studies proposing memorization measures avoided carefully examining the training dynamics of the model.” I think that this inaccurately describes the field. Other memorization papers have certainly looked at learning dynamics. The two that immediately spring to mind are Biderman et al. 2023 Emergent and Predictable Memorization in Large Language Models and Duan et al. 2025 Uncovering Latent Memories in Large Language Models.
- Line 149-150 “We choose formal grammar based languages because they can be fully learned without any memorization.” I think Strobl et al. 2024 What Formal Languages Can Transformers Express? A Survey and Bhattamishra et al. 2020 On the Ability and Limitations of Transformers to Recognize Formal Languages show that there are grammar-based languages that cannot be learned. Unless we’re considering a restricted subset of languages, I think this statement is incorrect.

## Appendix E

- Batch size is typically given in the number of tokens, and so “batch size is 8” is difficult to interpret; presumably this means that the number of sequences per optimizer step is 8? But what is the sequence length?
- If the tokens per optimizer step is not constant, how does it vary?
- If the number of epochs is held fixed but the training dataset sizes are varied, then these comparisons are not isoFLOP; is that correct? If so, it’ll be tricky to know whether whatever differences are found are indeed attributable to the experimental conditions or to other factors such as overtraining and optimization.
- How is tokenization handled? Do you extend the vocabulary with new unique tokens for the grammar? Or do you use the model’s tokenizers for each language without adapting it?

## Section 3

- Lines 185-186: I haven’t seen previous papers use cross entropy loss as a measure of memorization, and the cited Mao et al. 2023 does not mention memorization (as best as I can tell). I think it’d be good to use more “standard” measure(s) of memorization. See Section 2 of Xiong et al. 2025 The Landscape of Memorization in LLMs for a list of commonly used memorization metrics (no affiliation with the paper - I just thought they provide a nice comprehensive list of various memorization metrics in the literature)
- Equation 2: **At this moment, I realize that I have misunderstood what contextual memorization means.** To me, the mental model I had in my head was something along the lines: If the model’s context is string s = “John Smith’s email is X. What is John Smith’s email? ”, then if the model greedily decodes X, this would qualify as memorization but it isn’t really because the model is just using its context. Note: this doesn’t necessarily mean the model was or was not trained on s. The definition of contextual memorization in  equation (2) is quite different from this. It instead says: what is the smallest difference in loss between a model trained on this datum and models not trained on this datum.
- **I’m now very confused what “context” means in contextual memorization. Does it have any connection to the context window of a model? If not, what is the context in contextual memorization?**
- I now realize that Contextual memorization is a very slight variant on Counterfactual memorization, merely adding a $\min_e$
- In Section 3.2, the second and third findings (lines 273 and 282, respectively) seem like thinking in a vacuum (I wanted to say intellectual masturbation but am trying to elevate my language). What is the importance of these two findings? Why should anyone care? It seems like you’ve proposed a new metric that is a slight variation of a previous metric I’ve never heard of, and are going on about how these two metrics are related to one another without drawing any substantial conclusions.
- The takeaway statement on line 280 seems trivially obvious: “Different measures can disagree on the start and order of memorization.” If they didn’t disagree, then why would the field have like 10+ metrics for memorization?

- **Disclaimer: At this point, I’m pretty checked out of this paper. It comes across as much ado about nothing. If there’s a reason why this matters, conceptually or empirically, I don’t feel like I’ve encountered it yet.**

## Section 4

- Line 337 to 340: “Since contextual and counterfactual memorization are related (Lemma 1), we henceforth compare between contextual and recollection-based memorization.” gave me whiplash. We spent the first 6 pages making a hullabaloo about contextual vs counterfactual memorization, and then say “They’re basically super similar, so let’s just omit one.” It undermines the paper’s position that these metrics are meaningfully different.

## Section 6

- The transition to RQ6 is abrupt. I don’t know how this fits into the overall narrative or earlier results.
- I don’t see mention of what defines “privacy-sensitive PII” or how the authors made the classification of whether memorized strings do (or do not) contain privacy-sensitive PII.
- The use of a proxy model (in this section, OLMo-1B) seems unjustified, theoretically or empirically. Sure, one _can_ do this, but whether it approximates contextual memorization (or how well) is not substantiated.

**Questions:**

- Please see Weaknesses.

- One additional aspect that was unclear to me: the Abstract and Introduction claim to use 18 LLMs across 6 model families. When I look at the figures and tables in the main text, I don't see any mention of the model families. Could you please clarify the relevance of this claim? I understand that Appendix E includes the model families (Qwen, Gemma, Llama 3 - without citations by the way) but I don't understand the role that these families play in the main text. This detail seems almost irrelevant?

---

> ### Author Response · Authors · 2025-11-21
>
> We thank the reviewer for their elaborate feedback, which gives us an idea of how the paper appears to a reader. We incorporate suggestions in the final version of the paper. We address specific questions below.
>
>
> **Contextual memorization: a minor variant of counterfactual memorization**. In terms of operationalization, contextual memorization may look similar to counterfactual memorization, where the former considers the lowest counterfactual loss as the threshold for memorization, compared to instantaneous counterfactual loss as the threshold in the counterfactual memorization.
>
> We however disagree with the reviewer and emphasize that the motivation of the two measures are different: contextual memorization captures learning risks vs. privacy risks in the counterfactual memorization. Unless there is a different motivation, one may not arrive at considering the lowest counterfactual loss as a threshold. We refer to our response to Reviewer **Qw3D** for elaboration.
>
> **Title**. The main claim in the title that learning is considered infeasible without memorization is not a subjective opinion. This is empirically observed (refer to **RQ3** in Section 4).
>
> **Additional references.** We thank the reviewer for additional references, which we incorporate in the final version of the paper.
>
> **In-context learning vs contextual memorization.** Contextual memorization is different from in-context learning. Contextual memorization relies on the observation that when the model learns a language (via pre-training or fine-tuning), it acquires *contextual* understanding of the underlying language, enabling it to generate new strings without memorization. In-context learning is a different mode of learning, where information is presented in the input prompt for the model answer – without any explicit training. If the reviewer has suggestions to rename “contextual memorization”, we would be glad to know.
>
> **Line 149-150**. We remove the sentence in the final version. The sentence is not correct. Refer to our answer to Reviewer **hoWX**.
>
> **Tokenization.** We consider character-level tokenization, where each character in the string is given a separate token.  For clarity on token statistics, refer to Reviewer **hoWX** and **Qw3D**.
>
> **Loss as a metric for recollection.** Cross entropy loss is a native metric of recollection, which is directly observable from LLM. We have already discussed the memorization metrics mentioned in Xiong et al. 2025, which fall under the category of recollection-based memorization and counterfactual memorization (see Section 1 and Appendix C).
>
> **Clarity on Section 6.** Section 6 studies whether privacy risks of memorization reported by earlier works are substantial or not. We manually review strings that are claimed to be memorized based on recollection, and find that the majority of the strings do not contain any sensitive information. In contrast, many strings are predictable, and can be explained away using contextual memorization -- we consider a proxy model for this evaluation. Please refer to answer to Reviewer **Qw3D**.
>
> **LLM Families.** We expand experiments to multiple families to understand if our observation generalizes to other LLMs, which indeed does. The citations of models are in the main paper (Line 165-167).

---

> ### Comment · Reviewer_w7jk · 2025-11-22
> **Level-Setting: What is the "context" in contextual memorization?**
>
> Hopefully we can treat this in an interactive manner! I want to make sure I'm understanding what you're trying to communicate, here and in your manuscript.
>
> To level set, in my original review, I realized I had misunderstood what “context” means in "contextual memorization". This made it difficult for me to understand your paper. I asked:
>
> > I’m now very confused what “context” means in contextual memorization. Does it have any connection to the context window of a model? If not, what is the context in contextual memorization?
>
> In your response, you wrote:
>
> > Contextual memorization relies on the observation that when the model learns a language (via pre-training or fine-tuning), it acquires contextual understanding of the underlying language, enabling it to generate new strings without memorization.
>
> Could you please clarify what "context" means in this setting? I didn't understand in the original paper nor in your response. The response seems to me like the word "contextual" is being repeated without any further clarification: "contextual memorization" is "acquiring contextual understanding".
>
>
> (At least to me, the "context" of a language model is a very widely used and accepted terminology, referring to the sequence of tokens provided before generation/sampling from the language model begins. If you want to break that association, I think you need to be very explicit in what "context" means in your paper)

---

> > ### Author Response · Authors · 2025-11-22
> >
> > Thanks for your reply.
> >
> > Let's revisit the thought experiment in the paper to explain "context" in contextual memorization (Line 60-69). Consider a German speaker and an English speaker trying to memorize a paragraph in German language. We can safely assume that the German speaker knows the syntax and semantics of German language better than the English speaker, who may not know German. In our paper, we introduce the term *context* to explain this phenomenon, where the German speaker has higher contextual understanding of the German language than the English speaker. Therefore, if both speakers *could* generate or recollect the same paragraph with an equal confidence (i.e., proficiency or probability), the German speaker requires less (*contextual*) memorization than the English speaker. Because, the German speaker can possibly recollect much of the paragraph from their contextual understanding of the language.
> >
> > Now relating it to LLMs, when the LLM is trained on a dataset sampled from an underlying (formal) language, the LLM acquires contextual understanding of the language, enabling its generalization. Therefore, we can expect the LLM to generate any *unseen* string to some extent due to contextual understanding, which we refer as contextual threshold. Contextual memorization begins when the training recollection of a training string is higher than its contextual threshold, specifically the *optimal* contextual threshold (i.e., the best recollection of the string when it is excluded from training).
> >
> > > (At least to me, the "context" of a language model is a very widely used and accepted terminology, referring to the sequence of tokens provided before generation/sampling from the language model begins. If you want to break that association, I think you need to be very explicit in what "context" means in your paper)
> >
> > Your reference of "context" is in fact **in-context learning**, where the "in-" part refers to tokens in the prompt preceding the LLMs' generation. Our reference of "context" does not contain "in-". Rather, "context" refers to the ability of an LLM in understanding the language and resulting in its generalization.
> >
> > We will clarify the term in the final version of the paper. Thanks for your careful observation.

---

> ### Comment · Reviewer_w7jk · 2025-11-22
> **Thank You For Clarifying What Contextual Means To You**
>
> Thank you for the quick turnaround! I appreciate you clarifying what contextual means to you.
>
> In my opinion, it seems like "contextual" is a very poor choice of terminology. Beyond conflicting with the notion of a language model's context or context window, "contextual" doesn't seem to precisely specify or accurately capture any notion, or add anything really.
>
> To demonstrate my point, let's delete "contextual" from your previous response and reread it:
>
> > Consider a German speaker and an English speaker trying to memorize a paragraph in German language. We can safely assume that the German speaker knows the syntax and semantics of German language better than the English speaker, who may not know German. The German speaker has higher understanding of the German language than the English speaker. Therefore, if both speakers could generate or recollect the same paragraph with an equal confidence (i.e., proficiency or probability), the German speaker requires less memorization than the English speaker. Because, the German speaker can possibly recollect much of the paragraph from their understanding of the language.
>
> > Now relating it to LLMs, when the LLM is trained on a dataset sampled from an underlying (formal) language, the LLM acquires understanding of the language, enabling its generalization. Therefore, we can expect the LLM to generate any unseen string to some extent due to understanding, which we refer as the generation threshold. Memorization begins when the training recollection of a training string is higher than its generation threshold, specifically the optimal generation threshold (i.e., the best recollection of the string when it is excluded from training).
>
> To me, this explanation makes much more sense without "contextual" than it does with "contextual"
>
> Let me now reread your manuscript and our previous discussion with my newfound understanding.

---

> ### Comment · Reviewer_w7jk · 2025-11-22
> **With My Newfound Understanding of "Contextual" , You Should Cite and Discuss "Language Models May Verbatim Complete Text They Were Not Explicitly Trained On"**
>
> With my newfound understanding of what "contextual" means, I think there's a highly relevant paper you should relate your paper to:
>
> https://arxiv.org/abs/2503.17514
>
> Disclaimer: I am not an author on this paper - I just think it is highly relevant
>
> This paper is specifically relevant because your point is, as I paraphrased below, "Memorization begins when the training recollection of a training string is higher than its generation threshold, specifically the optimal generation threshold."
>
> Your paper thus gives a quantitative notion of memorization (i.e., how likely the model is to produce a training string ABOVE what one would expect from training on the grammar but not seeing that string) that directly answers this paper's point that a more naive measure of memorization (does the model produce a specific string) is inadequate!

---

> ### Comment · Reviewer_w7jk · 2025-11-22
> **Title and RQ3: Learning w/o Memorization Considered Infeasible**
>
> Based on my newfound understand, I am retracing the manuscript and our conversation thus far.
>
> Originally, I did not understand the title because of how it was phrased. I did not know if this was a position you held or whether it merely alluded to someone else's position, or whether the paper would try to make this point.
>
> I think my confusion was due to the pass voice and use of "considered". A title like "Learning Without Memorizing Is Infeasible" is clear and objective, whereas "Learning Without Memorization Considered Infeasible" is subjective and passive.
>
> Now that I understand this, I more strongly believe the authors should cite Feldman (2019) https://arxiv.org/abs/1906.05271 and subsequent works. These papers specifically argue that learning without memorization is infeasible.
>
> Continuing along, in the authors' response, they clarified that this is a position that they are advocating and say they demonstrate this in RQ3. Rereading RQ3 in this light makes me think the evidence is quite weak for two reasons:
>
> 1. Figure 3 shows that for Recollection with a threshold of 0.2, the fraction of memorized strings by the optimal test-loss epoch is 0. This then demonstrates that, at least under certain definitions of memorization, memorization is NOT necessary for learning
>
> 2. Moreover, even if we focus on Contextual and Counterfactual memorization, the evidence shows that some positive amount of memorization has occurred at the epoch of the lowest test loss _in this setting_. This is _not_ the same as demonstrating (either mathematically or with rigorous experimentation) that memorization is _necessary_ for learning.

---

> > ### Author Response · Authors · 2025-11-23
> > **Revised version of the paper**
> >
> > We have updated the paper with additional references and clarification. Notable changes are in Figure 1 and Appendix C (in blue color).
> >
> > We respect the opinion on the terminology "contextual memorization". We still think "contextual" term precisely communicates our message, where memorization should account for how much contextual understanding the LLM has on the underlying language and its contribution to the overall recollection/generation. If we find an alternate term, we will append in the final version of the paper.
> >
> > > https://arxiv.org/abs/2503.17514
> >
> > We have incorporated your suggestion to the related work (Line 921-927).
> >
> > > Now that I understand this, I more strongly believe the authors should cite Feldman (2019) https://arxiv.org/abs/1906.05271 and subsequent works. These papers specifically argue that learning without memorization is infeasible.
> >
> > We cite Feldman's paper, and mention that our work empirically supports their theoretical analysis (Line 915-921).
> >
> >
> > > Figure 3 shows that for Recollection with a threshold of 0.2, the fraction of memorized strings by the optimal test-loss epoch is 0. This then demonstrates that, at least under certain definitions of memorization, memorization is NOT necessary for learning
> >
> > In Line 353-359, we critically discuss the conclusion derived from recollection-based memorization, where the choice of threshold is manual, and could be error-prone. For example, one can always set a threshold loss close to 0, and show that no amount of memorization is exhibited by the model, and vice versa. Contextual and counterfactual memorization can avoid the fallacy of recollection-based memorization, where the threshold depends on the learnability of the underlying language.
> >
> > > Moreover, even if we focus on Contextual and Counterfactual memorization, the evidence shows that some positive amount of memorization has occurred at the epoch of the lowest test loss in this setting. This is not the same as demonstrating (either mathematically or with rigorous experimentation) that memorization is necessary for learning.
> >
> > We report results based on our experiments, where we find evidence that memorization cannot be fully avoided when optimally learning a language. Intuitively, memorization is unavoidable at optimal learning, because the beginning of memorization of different strings has a spread: some strings are memorized earlier than others, and empirically the optimal learning, which occurs at a single epoch, often comes after some strings are already memorized. If the reviewer has a different phrasing to communicate the message, we are curious to know.

---

### Official Review · Reviewer_hoWX · 2025-11-02

**Soundness:** 1
**Presentation:** 2
**Contribution:** 1
**Rating:** 2
**Confidence:** 4

**Summary:**

This paper motivates a new notion of memorization LLMs which dissects rote memorization vs "learning" (i.e. generalization). Towards this, they introduce "contextual memorization" that measures minimum difference between loss on target string s when trained w/ and w/o it over multiple epochs (the min is taken of differences over epochs). The paper discusses several implications of this notion and its results.

**Strengths:**

- I think this is an important problem to dissect memorization from generalization
- The paper has covered most of relevant prior related work that I'm aware of.

**Weaknesses:**

See questions

**Questions:**

- How do you define what a string is? Does it have a minimum or maximum length? If not, a single token is valid entry? How does this translates into epoching?
- Line 148-150 "...We choose formal grammar based languages because they can be fully learned without any memorization." Can you unpack this? Is this true? Because that does not make sense to me, please share your justification and preferably a cite for this.
- What is the motivation of reduction (min) over multi epoching on a dataset? I would just call it training longer, since epoching is dataset sample size specific. It is unclear what entails single epoch in LLM training. The entire paper is too dependent on this and for me, an epoch over sample of target distribution (training data) is too arbitrary measure.
- RQ4 is not an RQ since it is already answered by cited work.
- The main takeaway from the paper is "learning is infeasible without memorization" uses "contextual memorization" to show this by "by differentiating between context-based recollection and memorization-based recollection". After reading the paper, I think the term memorization is super overloaded and it has made it difficult to follow the claim. It seems its kind of cat and mouse game. I'm unsure if this is writing that is ambigous or the claims made as well.

---

> ### Author Response · Authors · 2025-11-21
>
> We thank the reviewer for their feedback. We answer the questions below.
>
> **String & Epoch.** A string is a sample of the underlying probabilistic formal grammar (representative strings are in Figure 12). The length distribution of strings are in Figure 11, where strings in language $L_1$ and $L_2$ have a uniform length of 72 tokens, while for $L_3$ and $L_4$, the length is between 10 and 90 tokens. During repeated training (specifically, fine-tuning), an epoch indicates a full iteration over all training strings by the LLM. Therefore, the LLM is trained on the same amount of tokens after each epoch.
>
> We will remove **Line 148-150** in the revised version. We choose formal grammar-based languages, because they offer a controlled setup where we can ensure that learning and memorization are unaffected by prior training of the models, free from data contamination, and guided by a tunable string distribution – enabling precise comparisons of the memorization measures.
>
> **Why minimization of loss over epochs?** The confusion might be from what entails training progression in LLM training. In the LLM *pre-training*, training progression is often viewed in terms of tokens seen by the model. We however consider *fine-tuning*, where we measure training progression by a full iteration over the training strings, called epochs, following standard terminologies in machine learning.
>
> Anyway, our motivation is to find the best contextual recollection (i.e., the lowest counterfactual loss) of any string, which we consider as a threshold to compute contextual memorization. Since we report losses after each epoch, the minimization is applied over all epochs. Regardless of epochs or tokens seen so far, the key idea is to find the lowest counterfactual loss of a string after training finishes. In the final version, we clarify this.
>
> **RQ4.** The cited work (we suppose deduplication papers) indeed connects higher string-frequency with higher memorization using *recollection-based memorization measures*. We however demonstrate that this is *not necessarily true* for contextual and counterfactual memorization, where both frequency groups can have similar memorization. The distinction is therefore significant to report.
>
> **Main takeaway.** Our goal is to study measures for memorization in LLMs, not only to capture privacy risks – which has been done in the past – but also to capture learning risks. We propose contextual memorization to precisely capture learning risks, and empirically show that when the LLM learns a language optimally, memorization is unavoidable. In the final version, we improve upon the presentation of the paper.

---

### Meta-Review · Area_Chair_4dhs · 2026-01-07

**Summary:**

The submission proposes "contextual memorization" to distinguish between rote memorization and generalization. While reviewers agreed the problem space is important, the execution was flawed. The primary friction points were the terminology—specifically the confusing use of "contextual" which conflicts with standard LLM nomenclature—and the failure to convincingly differentiate the new metric from existing "counterfactual" measures. Despite a rigorous back-and-forth, particularly with Reviewer w7jk, the core claims regarding the necessity of memorization for learning were deemed unsubstantiated by the empirical data provided.

**Reviewer Concerns:**

Addressed Concerns

These are issues where the authors provided a factual answer or concession that technically resolved the reviewer's specific question, even if it didn't save the paper.

Definitions of "String" and "Epoch" (Reviewer hoWX): The reviewer was confused about how these terms applied to formal languages. The authors clarified that strings are samples from the probabilistic grammar and epochs are full iterations over these training strings.

Inaccurate claims about Formal Languages (Reviewer hoWX, w7jk): Both reviewers flagged the claim that "formal grammar-based languages can be fully learned without any memorization" as scientifically incorrect. The authors conceded this was an error and agreed to remove the sentence.

Definition of "Context" (Reviewer w7jk): The reviewer initially could not find a definition for "context" in "contextual memorization." Through the discussion, the authors clarified their definition (referring to a model's underlying understanding of language syntax/semantics vs. the input window).

Missing Model Family Details (Reviewer w7jk): The reviewer noted a discrepancy between the abstract's claim of "18 LLMs" and the main text. The authors clarified these details were in the Appendix and cited in the main text.

Outstanding Concerns
These are issues that remained significant roadblocks to acceptance even after the rebuttal.

Poor Terminology / "Contextual" Confusion (Reviewer w7jk): While the authors defined what they meant by "context," Reviewer w7jk strongly argued that the choice of the word "contextual" was poor because it directly conflicts with the ubiquitous term "context window" in LLMs. The reviewer felt the explanation actually made more sense without the word "contextual," suggesting the terminology itself was obfuscating the contribution.

Substantiation of "Infeasibility" Claim (Reviewer w7jk, Qw3D): The authors claim that "learning is infeasible without memorization." Reviewer w7jk pointed out that the data (Figure 3) actually showed instances of optimal test loss with zero memorization under certain thresholds, directly contradicting the paper's title and main thesis. The authors' defense—that "memorization cannot be fully avoided when optimally learning"—was seen as moving the goalposts and not supported by the rigorous evidence required for such a strong claim.

Incremental Contribution vs. Counterfactual Memorization (Reviewer Qw3D, w7jk): Both reviewers felt the new metric was a minor variant of "counterfactual memorization." The authors argued it was distinct because it focused on "learning risks" rather than "privacy risks," but Reviewer Qw3D explicitly stated they were not convinced that this distinction made the new measure better or necessary.

Proxy Model Justification (Reviewer w7jk, Qw3D): The use of a proxy model to evaluate privacy risks was questioned. The authors explained why they did it (feasibility), but did not provide the empirical validation requested to prove that the proxy was actually an accurate substitute.

**Reviewer Scores:**

Here is how the scores likely would have changed (or not) had the discussion continued or if all reviewers participated fully:

1. Reviewer hoWX (Current Score: 2)

Predicted Score: 2 (Unlikely to change)

Reasoning: This reviewer did not engage after their initial review. While the authors answered their factual questions about "strings" and "epochs," the reviewer's deeper concern was that "the term memorization is super overloaded" and the paper felt like a "cat and mouse game" of definitions. Since the subsequent discussion with other reviewers confirmed that the terminology was indeed confusing and the definitions shaky, hoWX would likely have felt validated in their initial assessment that the presentation was poor (Soundness: 1).

2. Reviewer w7jk (Current Score: 2)

Predicted Score: 2 (Solidified Rejection)

Reasoning: This reviewer did participate fully. In fact, their score likely hardened from a "soft reject" to a "hard reject" during the discussion. Initially, they were confused. After the authors clarified the definition of "context," the reviewer realized they fundamentally disagreed with the terminology and the interpretation of the results. They explicitly noted that the new understanding made them believe the authors should be citing different literature (Feldman) and that the evidence for the main claim was "weak." There is no path to a higher score here; the clarification revealed deeper flaws.

3. Reviewer Qw3D (Current Score: 2)

Predicted Score: 2 (Unlikely to change)

Reasoning: This reviewer returned at the very end to say, "I maintain my original rating." They acknowledged the authors' explanations but remained unconvinced that the new metric was useful. Crucially, they noted that the "structure and organization" issues required a "major revision," which is code for "reject and resubmit." Even with more discussion, a reviewer who believes the paper needs a total rewrite will rarely change their score to acceptance based on forum comments alone.

---

### Decision · Program_Chairs · 2026-01-26

Reject